# Tubeimosides are pan-coronavirus and filovirus inhibitors that can block their fusion protein binding to Niemann-Pick C1

Ilyas Khan[1], Sunan Li[1], Lihong Tao[1], Chong Wang [1], Bowei Ye[2], Huiyu Li [2], Xiaoyang Liu[1], Iqbal Ahmad[1], Wenqiang Su[1], Gongxun Zhong [1], Zhiyuan Wen [1], Jinliang Wang [1], Rong-Hong Hua [1], Ao Ma[2], Jie Liang [2], Xiao-Peng Wan[1]✉, Zhi-Gao Bu [1]✉ & Yong-Hui Zheng [3]✉

SARS-CoV-2 and filovirus enter cells via the cell surface angiotensin-converting enzyme 2 (ACE2) or the late-endosome Niemann-Pick C1 (NPC1) as a receptor. Here, we screened 974 natural compounds and identified Tubeimosides I, II, and III as pan-coronavirus and filovirus entry inhibitors that target NPC1. Using in-silico, biochemical, and genomic approaches, we provide evidence that NPC1 also binds SARS-CoV-2 spike (S) protein on the receptor-binding domain (RBD), which is blocked by Tubeimosides. Importantly, NPC1 strongly promotes productive SARS-CoV-2 entry, which we propose is due to its influence on fusion in late endosomes. The Tubeimosides' antiviral activity and NPC1 function are further confirmed by infection with SARS-CoV-2 variants of concern (VOC), SARS-CoV, and MERS-CoV. Thus, NPC1 is a critical entry co-factor for highly pathogenic human coronaviruses (HCoVs) in the late endosomes, and Tubeimosides hold promise as a new countermeasure for these HCoVs and filoviruses.

Severe Acute Respiratory Syndrome Coronavirus 2 (SARS-CoV-2) has emerged as the 7th member of human coronaviruses (HCoVs) that also consist of two other highly pathogenic viruses SARS-CoV and Middle East Respiratory Syndrome-CoV (MERS), and four less pathogenic viruses. SARS-CoV-2 is the etiological agent of the Coronavirus Disease 2019 (COVID-19) pandemic, which has caused extraordinary damage to public health and the global economy. Although SARS-CoV-2 vaccines effectively prevent COVID-19 illness and death and are widely used, their inherent mutagenicity and such massive human intervention make the virus evolve rapidly, resulting in the emergence of SARS-CoV-2 variants of concern (VOCs) and breakthrough infections[1]. Currently, there are very limited options to treat these infections. Remdesivir and Paxlovid are the only antivirals approved by the Food and Drug Administration (FDA) to treat COVID-19 and Lagevrio can

only be used under an Emergency Use Authorization (EUA). Thus, there is an urgent need to develop novel antivirals for SARS-CoV-2/ COVID-19, especially those with broad spectrum, to complement the current vaccines.

SARS-CoV-2 virions are built up by four structural proteins including nucleocapsid (N), membrane (M), envelope (E), and spike (S) proteins. S protein is a class I fusion protein that mediates viral entry and its frequent mutations make a significant contribution to VOC. It is expressed as a type I transmembrane polypeptide precursor $S_0$ that is cleaved by furin to generate two non-covalently associated subunits $S_1$ and $S_2$ in producer cells. $S_1$ is displayed on the virion surface and has a receptor binding domain (RBD), which binds the angiotensin-converting enzyme 2 (ACE2) receptor on the target cell surface. $S_2$ is anchored on the virion membrane and triggers membrane fusion via

[1]State Key Laboratory for Animal Disease Control and Prevention, Harbin Veterinary Research Institute, Chinese Academy of Agricultural Sciences, Harbin, China. [2]Center for Bioinformatics and Quantitative Biology, Richard and Loan Hill Department of Biomedical Engineering, The University of Illinois Chicago, Chicago, IL 60607, USA. [3]Department of Microbiology and Immunology, The University of Illinois Chicago, Chicago, IL 60612, USA. ✉e-mail: wanxiaopeng@caas.cn; buzhigao@caas.cn; zhengyh@uic.edu

its N-terminal fusion peptide. The RBD-ACE2 binding induces the $S_1/S_2$ conformation changes that expose $S_2$, resulting in $S_2$ cleavage at the $S_2'$ or its adjacent site and activation of the fusion peptide. This proteolytic process is accomplished by the transmembrane serine protease 2 (TMPRSS2) on the cell surface, or cathepsin (CTS)-L in late endosomes. Based on the localization of these two cellular proteases, it has been thought that SARS-CoV-2 enters cells via two completely different routes: a TMPRSS2-dependent direct fusion on the plasma membrane, or a CTS-L-dependent fusion in late endosomes[2,3]. However, it was also reported that the productive SARS-CoV-2 entry is determined by extracellular pH regardless of the TMPRSS2 expression[4].

Niemann-Pick C1 (NPC1) is a large, multi-pass, 1278-amino acid (aa) transmembrane protein in late endosomes[5]. NPC1 co-opts with the soluble NPC2 to export cholesterol from late endosomes for redistribution to the other cellular membranes. In addition, NPC1 serves as the intracellular receptor of filoviruses[6,7]. NPC1 has 13 transmembrane domains, 3 large luminal domains (A, C, I), and a cytoplasmic C-terminal tail. Domain A is the 1st luminal domain that binds cholesterol for intracellular trafficking, and domain C is the 2nd luminal domain that binds Ebola virus (EBOV) glycoprotein (GP)[8]. EBOV-GP is another class I fusion protein cleaved by CTS-L/CTS-B and binds the two protruding loops of NPC1 domain C (NPC1-C) to trigger membrane fusion[9]. Here, we identified Tubeimosides as pan-entry inhibitors of filoviruses and highly pathogenic HCoVs and demonstrate the important role of NPC1 in SARS-CoV-2 entry.

## Results

### Tubeimosides inhibit EBOV entry

We screened a library containing 974 plant-sourced natural compounds using HIV-1 luciferase (Luc)-reporter pseudoviruses used in our previous studies[10,11]. We expressed EBOV-GP, vesicular stomatitis virus (VSV)-glycoprotein (G), and HIV-1 envelope glycoprotein (Env) on these pseudovirions to model their entry and identify the entry inhibitors (Supplementary Fig. S1). Initially, Vero and TZM-bI cells were incubated with these compounds at 10 µM for one hour (h) and infected with pseudoviruses expressing EBOV-GP or HIV-1 Env for screening. A total of 19 "hits" were identified that exhibited various inhibitory activities against these two viruses (Supplementary Fig. S2). These 19 compounds were further tested against EBOV, HIV-1, and VSV at 1 µM, and 5 compounds were found to specifically inhibit EBOV (Fig. 1A). These 5 compounds include Toremifene (Tor) and Tamoxifen (Tam) (Fig. 1B) that were already reported as EBOV entry inhibitors[12,13], which validated our screening assay. The other 3 compounds were Tubeimosides (Tubs) I, II, and III that are produced from *Bulbus Bolbostemma Paniculatum* (Fig. 1B). These five compounds were further tested at 0.5 µM using the pseudoviruses in Vero cells or using EBOV replication and transcription-competent virus-like particles (trVLPs) in HEK293T cells as we did previously[14,15]. From both tests, Tubs I/II/III reached almost 100% inhibition, whereas Tam and Tor reached only ~50%, indicating that Tubeimosides have a much more potent antiviral activity than Tam and Tor (Fig. 1C, left, middle). To further validate these results, we used HIV-1 pseudoviruses expressing green fluorescent protein (GFP)[10]. All these five compounds strongly reduced the number of GFP-positive cells when EBOV-GP, but not VSV-G, was used (Fig. 1C, right; Fig. 1D). However, to reach a similar level of inhibition, Tor and Tam were used at 5- or 10-fold higher concentrations (2.5 µM, 5.0 µM) than Tubs I/II/III (0.5 µM).

We then measured the anti-EBOV 50% maximal inhibitory concentration (IC$_{50}$) of these compounds in Vero, SNB-19, Huh-7, and HEK293T cells using pseudoviruses and/or trVLPs (Fig. 1E; Fig. 1F). U18666A was reported to inhibit EBOV entry[7], so its activity was also tested (Fig. 1B). The IC$_{50}$ values of U18666A, Tor, and Tam were much higher than Tubeimosides, and among Tubeimosides, Tub III showed the lowest IC$_{50}$ (~50 nM) (Fig. 1G). We further tested the cytotoxicity of Tubs I/II/III and found that their 50% maximal cytotoxic concentration

(CC$_{50}$) was ~10 µM (Fig. 1H). Collectively, we identified Tubeimosdies as novel EBOV entry inhibitors effective at nanomole concentrations, of which Tub III shows the strongest antiviral activity.

### Tubeimosides inhibit NPC1

To understand how Tubeimosides inhibit EBOV entry, we tested whether they block the receptor binding. Initially, we performed molecular docking and structural analysis via Webina (https://durrantlab.pitt.edu/webina/), a web server based on Autodock vina[16,17]. Based on the co-crystal structure of NPC1-C and EBOV-GP$_1$ (PDB: 5F1B)[9], NPC1-C has two protruding loops that are engaged with a large hydrophobic cavity in the RBD of GP$_1$ in the interactive interface of these two proteins (Fig. 2A). Tubs I/II/III were docked to Loop 1, which completely mask this domain that otherwise should bind RBD. Although Tam, Tor, and U18666A were also docked closely to Loop 1, their sizes were too small to mask Loop 1, which may explain their less potent anti-EBOV activity than Tubeimosides.

We then directly tested whether Tubeimosides inhibit the NPC1 activity by filipin staining, which detects unesterified cholesterol in late endosomes. Because U18666A, Tam, and Tor were reported to inhibit NPC1[18,19], they were included as controls. Tam, Tor, U18666A, and Tubs I/II/III all inhibited the NPC1 activity in SNB-19 cells in a dose-dependent manner (Fig. 2B). However, unlike Tor and Tam, the inhibition by U18666A and Tubs I/II/III were still detectable at 0.05 µM, indicating that they have a stronger activity than Tam and Tor. Because the Marburg virus (MARV) entry is also dependent on NPC1[6], we tested whether it is blocked by Tubeimosides. In fact, Tubs I/II/III blocked MARV entry up to 20- to 30-fold at 0.5 µM (Fig. 2C). These results confirm that Tubeimosides block filovirus entry by interfering with the function of NPC1.

Next, we carried out additional experiments to confirm that Tubeimosides indeed inhibit EBOV entry at the step of receptor binding. First, we did Time-of-Drug-Addition assay to determine the inhibitory kinetics as reported by the others[20]. Vero cells were infected with HIV-1 pseudoviruses expressing EBOV-GP, and treated with Tor, Tam, and Tubs I/II/III at different time points: 1 h before infection (−1 h), during infection (0 h), and 2 and 12 h after infection (+2 h, +12 h). Treatment at −1 h, 0 h, or 2 h inhibited >80% of EBOV-GP pseudovirus infection, and no inhibition was detected at 12 h (Fig. 2D), confirming that they all inhibit EBOV entry. Second, we determined whether Tubeimosides block EBOV internalization. EIPA [5-(N-ethyl-N-isopropyl) amiloride], which blocks macropinocytosis for EBOV internalization[12], was used as a control. Vero cells were pretreated with EIPA or Tubs I/II/III and inoculated with GFP-labeled EBOV VLPs expressing EBOV-GP or VSV-G. Viral internalization was detected by flow cytometry. EIPA blocked EBOV, but not VSV internalization, and Tubs I/II/III did not show any effect (Fig. 2E). Thus, unlike EIPA, Tubs I/II/III do not inhibit EBOV internalization. Third, we tested whether Tubeimosides inhibit CTS-B and CTS-L, which promote EBOV entry by cleaving GP. E-64-d ethyl ester (EST), which is a pan-cysteine protease inhibitor, was used as a control. Unlike EST, none of Tubeimosides inhibited the CTS-B/L activity (Fig. 2F; Supplementary Fig. S3). Lastly, we tested whether Tubeimosides inhibit endosomal acidification, which is required for EBOV entry. Unlike ammonium chloride (NH$_4$Cl), none of Tubeimosides showed any inhibition (Fig. 2G). Collectively, these results demonstrate that Tubeimosides target NPC1 to block EBOV entry.

### Tubeimosides inhibit SARS2-CoV-2 (SARS2) entry

Initially, we interrogated whether EBOV and SARS2 interfere with each other during entry. Huh-7 cells were treated with SARS2-VLPs expressing its S proteins and infected with HIV-1 pseudoviruses expressing EBOV-GP. Alternatively, Huh-7 cells expressing human ACE2 (A) and TMPRSS2 (T) (Huh-7-A-T) were treated with EBOV-VLPs expressing its GP and infected with HIV-1 pseudoviruses expressing SARS2-S. SARS2-

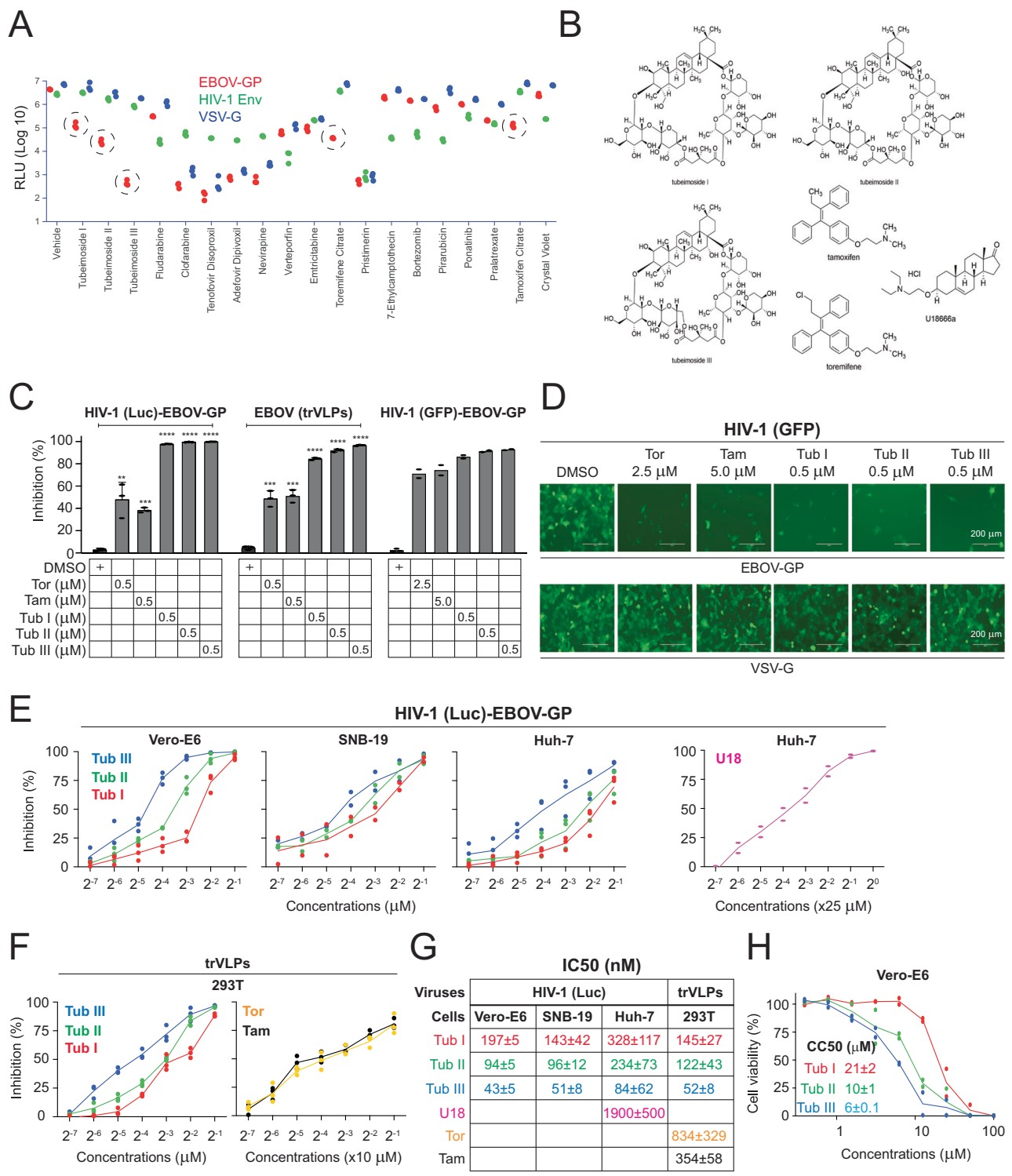

VLPs blocked EBOV entry and EBOV-VLPs blocked SARS2 entry in a dose-dependent manner (Fig. 3A), indicating that viral inference occurred during entry in certain cells. We also found that EBOV-VLPs expressing MARV-GP interfere with EBOV and SARS2, but not VSV entry (Supplementary Fig. S4). Thus, it is likely that SARS2 shares a similar entry pathway as filoviruses. Next, we tested whether Tubeimosides inhibit SARS2 infection using pseudoviruses. Indeed, Tubs I/II/III all blocked SARS2 infection, whereas Tor and Tam did not (Fig. 3B), and consistently, Tub III showed the lowest IC$_{50}$, albeit higher than the IC$_{50}$ recorded for inhibition of EBOV-GP pseudovirus infections (Fig. 3C). Time-of-Drug-Addition assay confirmed that

Tubeimosides block the SARS2 infection at the entry step (Fig. 3D). Thus, NPC1 may get involved in SARS2 entry.

We then tested the interaction between NPC1 and SARS2-S by immunoprecipitation (IP) followed by western blotting (WB). NPC1 pulled down EBOV-GP and SARS2-S, but not VSV-G (Fig. 3E), confirming the NPC1 binding to EBOV-GP and SARS2-S. To map the binding domain of NPC1, we created two NPC1 deletion mutants 1-377 and 1-620 that express domain A or domain A/C. Mutant 1-620 pulled down EBOV-GP and SARS2-S, whereas mutant 1-377 did not (Fig. 3F, lanes 5-8, 13-16), indicating that both EBOV-GP and SARS2-S bind NPC1-C. The NPC1 binding to EBOV-GP is promoted after GP is proteolytically

**Fig. 1 | Tubeimosides inhibit Ebola virus (EBOV) infection. A** A total of 19 compounds from a library containing 974 plant-sourced small molecules were tested at 1 μM for antiviral activity to EBOV, HIV-1, and VSV using HIV-1 Luc-reporter pseudoviruses. EBOV-GP (red)- and VSV-G (blue)-mediated infection were conducted in Vero-E6 cells, and HIV-1 Env (green)-mediated infection was done in TZM-bI cells. Five compounds specifically targeting EBOV are circled. Viral infection was shown as intracellular luciferase activity. RLU, relative light unit. **B** The chemical structures of six compounds are shown. **C** The anti-EBOV activity of Toremifene citrate (Tor), Tamoxifen citrate (Tam), and Tubeimosides (Tubs) I/II/III were determined at indicated concentrations using HIV-1 Luc-pseudoviruses, EBOV trVLPs, and HIV-1 GFP-pseudoviruses. Levels of viral infections were calculated as relative values, with the DMSO treatment set as 100. The percentage of inhibition was calculated by deduction of the relative viral infection value from 100. Error bars indicate SEMs

($n = 3$ biologically independent experiments). One-way ANOVA was applied. $**P < 0.01$, $***P < 0.001$, $****<0.0001$. **D** Vero-E6 cells were infected with HIV-1 GFP-pseudoviruses expressing EBOV-GP and treated with Tor, Tam, and Tubs I/II/III at indicated concentrations. Infected cells were imaged by EVOS FL Auto Imaging System (scale bar, 200 μm). **E** The anti-EBOV activity of Tubs I/II/III was titrated in Vero-E6, SNB-19, and Huh-7 cells using HIV-1 Luc-pseudoviruses. The anti-EBOV activity of U18666A (U18) was also titrated in Huh-7 cells similarly. The percentage of inhibition was calculated similarly as aforementioned. **F** The anti-EBOV activity of Tam, Tor, and Tubs I/II/III was titrated in HEK293T cells using EBOV trVLPs. The percentage of inhibition was calculated similarly as aforementioned. **G** A summary of $IC_{50}$ values on viral infection. **H** The cytotoxic effect of Tubs I/II/III was titrated in Vero-E6 cells using CellTiter-Glo® Luminescent Cell Viability Assay. Their $CC_{50}$ values are shown.

cleaved[8]. We cleaved EBOV-GP and SARS2-S on pseudovirions by bacterial metalloprotease thermolysin (TL) (Fig. 3G), which mimics this proteolytic process during entry. We purified recombinant NPC1 proteins from HEK293T cells and tested NPC1 binding to these cleaved and uncleaved EBOV-GP and SARS2-S by pulldown. NPC1 pulled down the cleaved EBOV-GP much more efficiently than uncleaved proteins (Fig. 3H, lanes 1, 3). However, NPC1 pulled down both cleaved and uncleaved SARS2-S at a similar efficiency (Fig. 3H, lanes 5, 7).

Lastly, we determined how Tubeimosides affect NPC1 binding to EBOV-GP and SARS2-S. U18666A was recently reported as a SARS2 inhibitor[21]. We confirmed that U18666A blocks SARS2 entry using pesudoviruses expressing SARS2-S in Huh-7-A-T cells, but its $IC_{50}$ was ~10 to 30-fold higher (1,500 nM) than Tubeimosides (Fig. 3I). U18666A and Tubs I/II/III all blocked the NPC1 binding to cleaved EBOV-GP, and consistently, Tub III exhibited the strongest activity (Fig. 3J, lanes 1-10). In addition, these compounds blocked NPC1 binding to cleaved SARS2-S (Fig. 3J, lanes 11-23) and uncleaved SARS2-S (Fig. 3J, lanes 24-32), and again, U18666A and Tub III showed the weakest or the strongest activity, respectively. Thus, Tubeimosides inhibit EBOV and SARS2 entry by blocking their binding to NPC1.

## NPC1 is a critical factor for SARS2-S pseudovirus infection

To elucidate the role of NPC1 in SARS2 infection, we knocked out *NPC1* using lentiviral vectors expressing CRISPR/Cas9 and *NPC1*-specific short-guide RNAs. Chinese hamster ovary (CHO) (Supplementary Fig. S5), human adenocarcinomic alveolar basal epithelial A549 (Supplementary Fig. S6), human colon epithelial Caco2 (Supplementary Fig. S7), African green monkey kidney epithelial Vero (Supplementary Fig. S8), and human airway epithelial Calu3 cells were transduced with these vectors to generate *NPC1*-knockout (KO) clones. Multiple KO clones were isolated from these cell lines after puromycin selection, except for Calu3 cells which were very difficult to isolate single colonies. Thus, these Calu3 cells were used as *NPC1*-knockdown (KD) cells. Human *ACE2* (A) was stably expressed in CHO, A549, and Vero WT cells to support and promote SARS2 replication, and human *TMPRSS2* (T) was also stably expressed in Vero cells. Unlike A549 cells, both Calu3 and Caco2 cells express high levels of endogenous TMPRSS2[22]. *NPC1*-KO/KD and ectopic *ACE2* expression were confirmed by WB (Fig. 4A). The ectopic ACE2 expression on the cell surface of A549 cells was confirmed by flow cytometry (Fig. 4A). Detection of ectopic *TMPRSS2* expression by WB was unsuccessful, although its biological activity was confirmed (see below). We also confirmed that *NPC1* KO did not affect A549, Caco2, CHO, and Vero cell viability and growth (Supplementary Fig. S9).

We then used HIV-1 pseudoviruses expressing SARS2-S, EBOV-GP, or VSV-G to infect these cells. Initial experiments were conducted in CHO cells. VSV-G and EBOV-GP pseudoviruses productively infected CHO wild-type (WT) and WT-A cells; *NPC1*-KO did not affect VSV-G pseudovirus infection, but reduced EBOV-GP pseudovirus infection up to 1000-fold (Fig. 4B). SARS2-S pseudovirus only infected CHO WT-A cells, and *NPC1*-KO similarly reduced the SARS2-S pseudovirus

infection as EBOV-GP pseudovirus (Fig. 4B). Thus, NPC1 is critically involved in SARS2 infection of CHO cells.

NPC1 P691S is a defective mutant in cholesterol uptake, and L656F and D786N are NPC1 gain-of-function mutants[7]. When they were expressed in CHO KO-A cells, they all promoted EBOV-GP and SARS2-S pseudovirus infection as the WT NPC1 protein (Fig. 4C), indicating that the cholesterol transport activity is not required for SARS2 infection, as reported in EBOV infection[7].

We then used human cells and Vero cells to confirm the critical role of NPC1 in SARS2 infection. Initial experiments were conducted in A549 cells. As expected, *NPC1*-KO selectively disrupted EBOV-GP pseudovirus infection but not VSV-G pseudovirus infection in these cells (Fig. 4D). Because A549 WT-A cells cannot be infected with SARS2-S pseudoviruses, we transfected these cells with a human TMPRSS2 expression vector. Ectopic ACE2 plus TMPRSS2 expression strongly promoted SARS2-S pseudovirus infection only in the presence of *NPC1* (Fig. 4E). We then infected Caco2 cells and Vero WT-A-T cells with these different pseudoviruses. Productive EBOV-GP, VSV-G, and SARS2-S pseudovirus infection were detected in these cells; *NPC1*-KO did not affect VSV-G pseudovirus infection, but reduced EBOV-GP and SARS2-S pseudovirus infection (Fig. 4F).

## NPC1 is a critical factor for authentic SARS2 infection

We next used SARS2 authentic viruses to infect A549, Calu3, and Caco2 cells and detected the viral replication using real-time PCR after 24 or 48 h as we reported previously[23]. Authentic SARS2 did not infect A549 WT cells, but productively infected A549 WT-A, Calu3, and Caco2 cells (Fig. 4G), as reported by the others[24,25]. *NPC1*-KO reduced the viral RNA copies by 100- to 1000-fold in A549 WT-A and Caco2 cells, and *NPC1*-KD reduced the viral RNA copies by 10- to 20-fold in Calu3 cells. The less NPC1-dependency by *NPC1*-KD was likely due to incomplete *NPC1* silencing. We also knocked down *NPC1* in Type II human primary alveolar epithelial (HPAE II) cells by small interfering RNAs (siRNAs) and found that the authentic virus infection was inhibited by over 40-fold (Fig. 4H). We further confirmed the antiviral activity of Tubeimosides. When Vero cells were treated with Tubs I/II/III and infected with authentic SARS2, they inhibited the viral infection in a dose-dependent manner, with $IC_{50}$ of 350 nM, 100 nM, or 50 nM, respectively (Fig. 4I). Collectively, these results confirm that NPC1 plays a critical role in SARS2 infection.

## NPC1-C binds SARS2 RBD

To understand how NPC1-C interacts with SARS2-S, we used the ColabFold[26–28] server version of the AlphaFold2-Multimer (AFM)[29], to predict the protein-protein complex of NPC1-C (PDB: 5HNS)[30] and SARS2-S protein (UNIPROT: P0DTC2)[31]. The SARS2-S spans residues 1 to 1273, and the NPC1-C is represented by residues 1274 to 1517. The top-ranked prediction located the NPC1-C-binding region to RBD that spans residues 340 to 512 (Fig. 5A; Supplementary Fig. S10). Both NPC1-C and SARS2-S exhibit a reliable depth of multiple sequence alignment, surpassing 100 (Fig. 5B). The predicted local-distance difference test

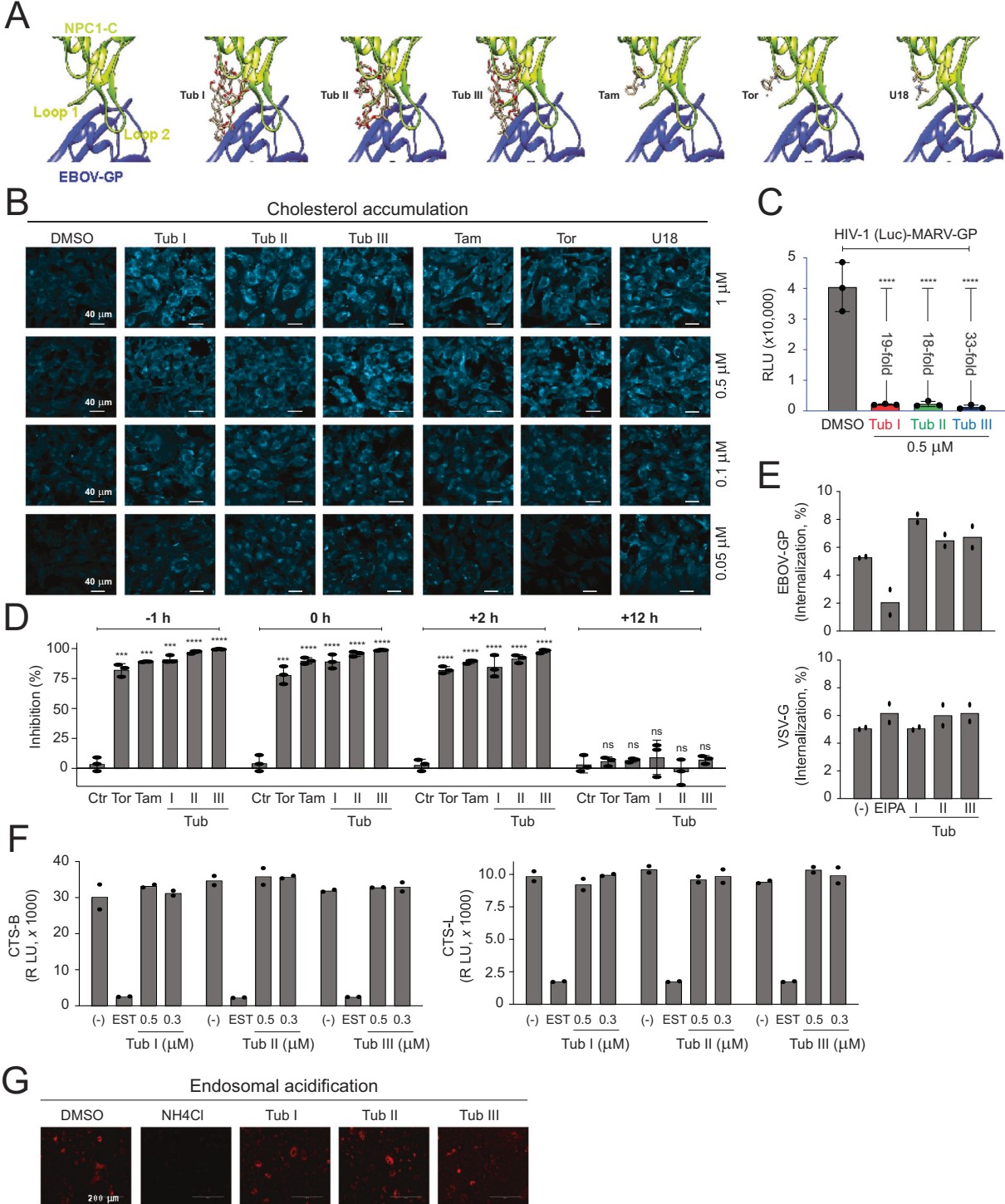

**Fig. 2 | Tubeimosides inhibit NPC1. A** Docking of indicated compounds to NPC1-C and EBOV-GP complex (PDB: 5F1B) was analyzed via Webina. **B** SNB-19 cells were treated overnight with indicated compounds. After being fixed, cells were stained with filipin (50 μg/mL) and visualized by confocal microscopy (scale bar, 40 μm). Experiments were repeated 3 times independently, and representative results are shown. **C** Vero-E6 cells were treated with Tubs I/II/III at 0.5 μM and infected with HIV-1 Luc-pseudovirus expressing MARV-GP. Viral infection was determined as previously. **D** Vero-E6 cells were treated with indicated compounds and infected with HIV-1 Luc-pseudovirus expressing EBOV-GP. Tor, Tam, and Tubs I/II/III were used at 2.5 μM, 5.0 μM, or 0.5 μM, respectively. DMSO was used as a control (Ctr). **E** Vero-E6 cells were pretreated for 1 h with EIPA and Tubs I/II/III and spinoculated with GFP-labeled EBOV-VLPs expressing EBOV-GP or VSV-G. After removal of VLPs and culture for another 3 h, GFP-positive cells were quantified by flow cytometry. EIPA [5-(N-ethyl-N-isopropyl) amiloride] and Tubs I/II/III were used at 25 μM, or 0.5 μM, respectively. **F** CTS-B or CTS-L substrates were incubated with cell lysate from A549 cells treated with Tubs I/II/III, and their activity was determined. E-64-d ethyl ester (EST) was used at 10 μM and Tubs I/II/III were used at indicated concentrations. **G** A549 cells were treated with NH₄Cl at 100 μM and Tubs I/II/III at 0.5 μM. After being stained with LysoTracker Red, cells were visualized by fluorescence microscope (EVOS FL Auto Imaging System) (scale bar, 200 μm). Experiments were repeated 3 times independently, and representative results are shown. Error bars in **C** and **D** indicate SEMs (*n* = 3 biologically independent experiments). One-way ANOVA was applied. ***$P < 0.001$, ****$P < 0.0001$; n.s. not significant.

(pLDDT) scores for this interacting region of SARS2-S (residue 340-520) and the entire structure of NPC1-C were between 70 to 90, indicating high levels of confidence in predictions of these residues (Fig. 5B). The details of top-ranked SARS2-S binding surface residues are summarized in Supplementary Table S1.

To further validate the specific interaction between NPC1-C and SARS2-S, we repeated the NPC1 and SARS2-S pulldown assay in the presence of recombinant soluble ACE2 proteins. ACE2 inhibited SARS2-S pseudovirus infection in a dose-dependent manner (Fig. 5C), confirming its biological activity. Thermolysin-digested pseudovirions expressing SARS2-S were incubated with increasing amounts of ACE2, followed by another incubation with purified NPC1, and proteins were immunoprecipitated with anti-NPC1 or SARS2-S. The pulldown of SARS2-S by NPC1 (Fig. 5D, lanes 2-4) and NPC1 by SARS2-S (Fig. 5D, lanes 6-8) were blocked by ACE2 in a dose-dependent manner, suggesting that NPC1-C binds SARS2-S on RBD. To further confirm this interaction, we used purified recombinant NPC1-C and SARS2-S RBD proteins to detect their direct interaction by ELISA (Fig. 5E). Not only the interaction was detected in a NPC1-C dose-dependent manner (Fig. 5F), but also this interaction was blocked by Tub III in a dose-dependent manner (Fig. 5G). Collectively, these results demonstrate that NPC1-C binds SARS2-S on RBD.

### SARS2 entry is blocked by lysosomotropic agents

Although it was assumed that low pH is only required for CTS-L mediated SARS2 entry, a recent study showed that mildly low pH is also required for the TMPRSS2-mediated entry[4]. Because NPC1 is a critical factor for SARS2 infection of TMPRSS2$^+$ (Caco2, Calu3) cells (Fig. 4G), we determined how the acidic environment affects SARS2 infection in TMPRSS2$^+$ cells.

Huh-7-A-T and Caco2 cells were treated with increasing concentrations of chloroquine and ammonium chloride and infected with HIV-1 pseudoviruses expressing SARS2-S, EBOV-GP, or VSV-G. In addition, TZM-bI cells were treated similarly and infected with pseudoviruses expressing HIV-1 Env. Both weak bases blocked the SARS2, EBOV, and VSV entry in a dose-dependent manner in both cell lines but had little effect on HIV-1 entry in TZM-bI cells (Fig. 5H). However, VSV and EBOV entry are more sensitive to these lysosomotropic agents than SARS2 entry. These results confirm that mildly low pH is required for SARS2 entry in TMPRSS2$^+$ cells.

### NPC1 is a critical factor for SARS-CoV (SARS1), MERS-CoV (MERS), and SARS2 VOC infection

To understand the general role of NPC1 in HCoV infection, we interrogated whether NPC1 is required for SARS1 and MERS entry. Initially, Huh-7-A-T cells were treated with Tor, Tam, and Tubs I/II/III at 1 μM and infected with pseudoviruses expressing SARS1-S or MERS-S. Tubs I/II/III inhibited both SARS1-S and MERS-S pseudovirus infection more than 75%, whereas Tor and Tam did not have any activity (Fig. 6A). Next, NPC1-KO cell lines were used for infection with these pseudoviruses. SARS1-S and MERS-S pseudoviruses productively infected Caco2 WT cells and Vero WT-A-T cells, and NPC1-KO strongly reduced these infections (Fig. 6B). We further tested SARS1-S and MERS-S interaction with NPC1 by IP as we did previously. NPC1 pulled down SARS1-S and MERS-S but did not pull-down influenza virus haemagglutinin (HA) from the A/WSN/1933 (H1N1) strain (Fig. 6C).

Next, we tested whether SARS2 VOCs are still dependent on NPC1 for entry. We produced HIV-1 pseudoviruses expressing S proteins from VOCs including Alpha, Beta, Gamma, Delta, and Omicron (Supplementary Fig. S11). We also tested S proteins from Lambda, Kappa, and the D614G mutant. When Caco2 cells were infected with these pseudoviruses, NPC1-KO reduced their infectivity by ~1000-fold (Fig. 6D). When Huh-7-A-T cells were infected with these pseudoviruses and treated with Tubeimosides at 0.5 μM, their infectivity was all inhibited to a similar level, although Kappa and Lambda were

slightly less sensitive (Fig. 6E). We further tested their interaction with NPC1 by IP and found that S proteins from D614G, Omicron, and Delta still interacted with NPC1 and the 1-620 mutant, but they did not interact with the 1-377 mutant except the Omicron S protein (Supplementary Fig. S12).

We conducted similar in-silico analyses to confirm these results using the Omicron S protein structure (PDB: 7WG7)[32]. It was predicted that NPC1-C also binds Omicron S RBD (Fig. 6F), which has a similar reliable multiple sequence alignment depth and high pLDDT scores (Supplementary Fig. S13). In addition, Tubs I/II/III were also docked to the interface of NPC-1-C and Omicron S RBD (Fig. 6G), but Tor, Tam, and U18666A were not.

To prove the specificity of these experiments, we infected A549 cells with influenza A viruses, which also express class I fusion protein HA. Tubs I/II/III did not inhibit the viral infection (Fig. 6H), and NPC1 was not required for the infection (Fig. 6I). In addition, when exogenous cholesterol was added during SARS2-S pseudovirus infection of Caco2, although it slightly increased the viral infection in the WT cells, it did not rescue the viral infection in NPC1-KO cells (Supplementary Fig. S14). Thus, NPC1 should not act indirectly via cholesterol to promote SARS2 infection.

### Discussion

NPC1 has been implicated to play a role in SARS-CoV-2 infection. Initial investigation in monkey kidney cells showed that SARS-CoV shares the same late entry kinetics with EBOV by entering NPC1-positive late endosomes[33]. Based on the role of cholesterol and late endosomes in SARS-CoV-2 infection, NPC1 was proposed as a therapeutic target for COVID-19 during the early time of this pandemic[34,35]. NPC1 was identified as an indispensable host factor for SARS-CoV-2 infection from two independent genome-wide CRISPR screening[36,37]. U18666A, an EBOV entry inhibitor targeting NPC1[7], was also found to inhibit SARS-CoV-2 infection[21,38,39]. However, when SARS-CoV-2 entry was determined in NPC1-KO HEK293T-ACE2-TMPRSS2 cells, the reduction of viral infection was only around 3-fold[21]. On the other hand, it was concluded that NPC1 interacts with the viral N protein to promote SARS-CoV-2 infection[38]. Now, we have compelling evidence that NPC1 is involved in SARS-CoV-2 entry. We identified Tubeimosides as novel pan-entry inhibitors of filoviruses and highly pathogenic HCoVs that target NPC1. Using CRISPR/Cas9 and siRNAs, we demonstrate that NPC1 is a critical factor for SARS-CoV-2 infection of different cell types from different tissues/species, including HPAE II cells and the human lung cell lines A549 and Calu3. Like HPAE II cells, A549 cells are alveolar epithelial type 2 cells (AT2s)[40], which are infected by SARS-CoV-2 to cause lung injury and impaired air exchange in COVID-19 patients. We also found that the infection of SARS-CoV-2 VOCs, SARS-CoV, and MERS-CoV are dependent on NPC1. Thus, NPC1 is a critical entry cofactor for these HCoVs.

Using high-resolution live-cell 3D imaging, a recent study has found that SARS-CoV-2 fusion with cell membrane requires acidic pH and uncovered three productive entry routes[4]. When cells are cultured at a neutral pH of 7.4, viruses enter cells via NPC1-positive late endosomes (pH ≤ 5.5 to 6) in TMPRSS2$^-$ cells or via these late endosomes and NPC1-negative early endosomes (pH ~6.0 to 6.8) in TSPRSS2$^+$ cells. When TMPRSS2$^+$ cells are cultured at a mildly acidic pH of 6.8, viruses enter cells via the plasma membrane and late endosomes, which may occur in nasal cavity during initial clinical infection. Thus, SARS-CoV-2 may enter cells via later endosomes, early endosomes, and/or the plasma membrane depending on the extracellular pH and TMPRSS2 expression (Fig. 7). We found that the VSV and EBOV entry are more sensitive to the lysosomotropic agents than SARS-CoV-2, which supports that unlike VSV and EBOV, SARS-CoV-2 could use these alternative routes for infection. We also observed that unlike in

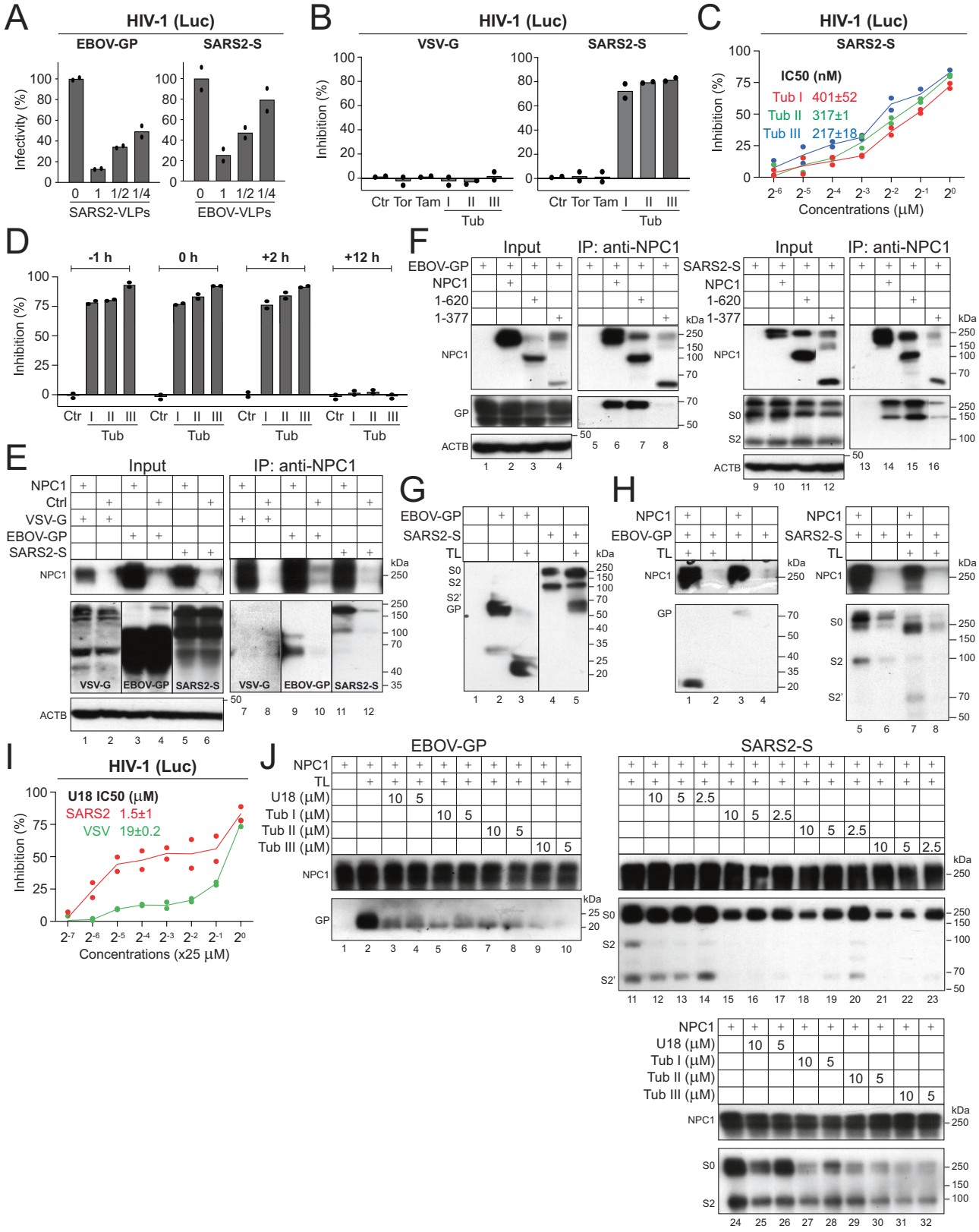

Vero cells, *NPC1*-KO does not inhibit SARS-CoV-2 entry as completely as it did for EBOV in CHO and Caco2 cells. These results suggest that the choice of SARS-CoV-2 entry routes could be cell type-dependent. Nonetheless, NPC1 should play an indispensable role in SARS-CoV-2 entry through late endosomes. Very recently, TMEM106B, another late endosome protein, was identified as an

ACE2-independent SARS-CoV-2 receptor[41,42]. Thus, the role of NPC1 as a SARS-CoV-2 intracellular receptor should be further explored.

Our in-silico and biochemical analysis demonstrate that NPC1-C binds SARS-CoV-2 RBD as ACE2 does, which opens an interesting question on the role of ACE2 and NPC1 in SARS-CoV-2 entry through

**Fig. 3 | Tubeimosides inhibit SARS2 entry. A** Huh-7 cells were treated with SARS2-VLPs and infected with HIV-1 Luc-pseudovirus expressing EBOV-GP. Alternatively, Huh-7-A-T cells were treated with EBOV-VLPs and infected with HIV-1 Luc-pseudovirus expressing SARS2-S. Viral infection is presented as relative values, with the infection in the absence of VLPs set as 100. **B** Huh-7-A-T cells were infected with HIV-1 Luc-pseudoviruses expressing VSV-G or SARS2-S and treated with indicated compounds at 1 μM. DMSO was used as a control (Ctr). The percentage of inhibition was calculated similarly as aforementioned. **C** The anti-SARS2 activity of Tubs I/II/III were measured in Huh-7-A-T cells using HIV-1 pseudovirus expressing SARS2-S, and their $IC_{50}$ values are indicated. **D** Huh-7-A-T cells were treated with 1 μM Tubs I/II/III at indicated time points and infected with HIV-1 Luc-pseudovirus expressing SARS2-S. DMSO was used as a control (Ctr). Viral inhibition was determined as previously, **E** NPC1 was expressed with EBOV-GP, VSV-G, or SARS2-S in HEK293T cells. NPC1 was immunoprecipitated (IP), and proteins in cell lysate (input) and pulldown samples were analyzed by WB. **F** NPC1 and deletion mutants 1-620 or 1-377 were expressed with EBOV-GP or SARS2-S in HEK293T cells. These NPC1 proteins were immunoprecipitated and proteins were detected by WB. **G** HIV-1 Luc-pseudoviruses expressing EBOV-GP or SARS2-S were purified by ultracentrifugation. After treatment with thermolysin (TL) at 200 μg/mL, proteins were analyzed by WB. **H** Recombinant NPC1 proteins were purified by immunoprecipitation from HEK293T cells transfected with a NPC1-expression vector. NPC1 was incubated with purified HIV-1 pseudoviruses expressing EBOV-GP or SARS2-S pre-treated with TL. Proteins associated with NPC1 were pulled down and analyzed by WB. **I** The anti-SARS2 activity of U18666A (U18) was measured in Huh-7-A-T cells using HIV-1 Luc-pseudovirus expressing SARS2-S. **J** Purified recombinant NPC1 proteins were incubated with indicated compounds, followed by incubation with HIV-1 pseudoviruses expressing EBOV-GP or SARS2-S after cleavage by TL. Proteins associated with NPC1 were pulled down and analyzed by WB. Experiments in **E**–**J** were repeated 3 times independently, and representative results are shown.

late endosomes. It was reported that the endogenous ACE2 proteins are present as monomers on the plasma membrane at low densities, and on average, only one S protein per virion can bind one ACE2 simultaneously[43]. In addition, when recombinant S trimers and ACE2 proteins were incubated at 1:2 ratio, ~40% of S trimers were not associated with ACE2, and ~75% of the S protein-ACE2 complexes accommodated only one ACE2[44]. Thus, there are many free RBDs available to NPC1 when viruses enter late endosomes. Using a biochemical assay, it was shown that ACE2 is not required for fusion when an alternate means of attaching SARS-CoV-2-S particles to target membranes was provided[45]. From these findings, we suggest that in some cells, ACE2 is more likely an attachment factor for SARS-CoV-2 internalization. After entering late endosomes, SARS-CoV-2 is engaged with NPC1 for membrane fusion (Fig. 7). From this regard, ACE2 should play a similar role as T-cell immunoglobulin and mucin domain (TIM) proteins, which serve as attachment factors for EBOV to initiate endocytic entry[46]. However, there is a remarkable difference between ACE2 and these attachment factors. ACE2 binds the viral RBD and promotes the next step of proteolytic cleavages, whereas TIM proteins do not. Thus, future work is required to visualize the SARS-CoV-2-S and NPC1 interaction and to show that it is functionally relevant to fusion.

Of six NPC1 inhibitors tested, they all blocked EBOV entry, but only Tubeimosides strongly inhibited SARS-CoV-2 infection. Tub I was identified as an anti-SARS-CoV-2 agent from a previous study[47]. Tub I showed very slow clearance, with maximum concentration ~1000 nM at ~2.85 h in rat plasma after intravenous administration[48]. Of three Tubeimosides, Tub II, and in particular Tub III, strongly inhibit EBOV and SARS-CoV-2 infection at ~50 to 100 nM levels. Thus, Tubeimosides hold promise as novel antiviral compounds for filovirus and highly pathogenic HCoVs.

## Methods
### Chemical reagents and recombinant proteins
Tubeimosides I (Cas No. 102040-03-9), Tubeimosides II (Cas No. 115810-12-3), and Tubeimosides III (Cas No. 115810-13-4) were purchased from Chengdu Biopurity Phytochemicals Ltd, China; U18666A (Cat No. 50205-0551) was from Cell Signaling Technology; Cathepsin B substrate (Cat No. sc-215529) and Cathepsin L substrate (Cat No. sc-3136) were from Santa Cruz Biotechnology; Filipin III (Cat No. GC12048) and CA074 (Cat No. GC15917) were from GLPBIO; EIPA (Cat No. HY-101840) and EST (E-64-d) (Cat No. HY-100229), and cholesterol (Cat No. HY-N0322) were from MedChemExpress; N-Dodecyl-β-D-maltoside (DDM) (Cat No. D4641), recombinant soluble ACE2 (Cat No. SAE0064), thermolysin from *Bacillus thermoproteolyticus* (Cat No. P1512), and anti-protease inhibitor cocktail (Cat No. P8340) were from Sigma Aldrich; Lyso-Tracker Red (Cat No. C1046), phosphoramidon (Metalloproteinase inhibitor) (Cat No. SG2024), and acetate buffer pH5.2 (Cat No. ST351) were from Beyotime Biotechnology; Lipofectamine 3000 (Cat No. L3000015) and Blasticidin S HCl (R21001) were from ThermoFisher;

Puromycin (Cat No. ant-pr-1) was from InvivoGen; polyethyleneimine (Cat No. 23966-1) was from Polysciences, Inc; recombinant NPC1-C protein (Cat No. 16499-H32H) and recombinant SARS-CoV-2 spike RBD protein (Cat. No. 40592-V08H) were from SinoBiological.

### Antibodies
Goat anti-human ACE2 affinity purified IgG (Cat No. AF933) (1:5000) was from R&D Systems; rabbit anti-human NPC1 (Cat No. ab134113) (1:5000) was from Abcam; rabbit anti-SARS-CoV-spike protein (Cat No. NB100-56047) (1:3000) was from Novus Biologicals; rabbit anti-SARS-CoV-2 spike RBD protein (Cat No. 40592-T62) (1:3000), rabbit anti-Zaire EBOV-GP (Cat No. 40442-T48) (1:5000) were from Sino Biological; mouse monoclonal anti-VSV-G (Cat No. Cat#A02180) (1:2000) was from Abbkine; rat monoclonal anti-FLAG (Cat No. F3165) (1:10,000), horseradish Peroxidase (HRP)-conjugated anti-FLAG (Cat No. A8592) (1:10,000), HRP-conjugated anti-HA (Cat No. H6533) (1:10,000), and HRP-conjugated anti-β-Actin (Cat No. A3854) (1:5000)were from Sigma Aldrich; HRP-conjugated AffiniPure donkey anti-goat IgG (H + L) (Cat. No. 705-035-003), HRP-conjugated goat anti-mouse IgG (Cat No. 115-035-003) (1:5000) and anti-rabbit IgG (Cat No. 111-035-003) (1:5000), and APC-conjugated polyclonal anti-goat antibody (Cat No. 705-136-147) (1:500) were from Jackson ImmunoResearch.

### Cells
HEK293T (Cat No. CRL-3216), Caco-2 (Cat No. HTB-37), Calu-3 (Cat No. HTB-55), Vero-E6 (Cat No. CRL-1586), SNB-19 (Cat No. CRL-2219), A549 (Cat No. CCL-185), and CHO-K1 (Cat No. CCL-61) cell lines were purchased from American Type Culture Collection (ATCC). TZM-bI cell line (Cat No. ARP-8129) was obtained from NIH HIV Reagent Program. Huh-7 cell line (Cat No. 1101HUM-PUMC000679) was purchased from Institute of Basic Medical Sciences (IBMS), Chinese Academy of Medical Sciences (CAMS) & Peking Union Medical College (PUMC). All these cells were cultured in Dulbecco's modified Eagle's medium (DMEM) with 10% fetal bovine serum (FBS) and 1% pen/strep under controlled conditions of incubation at 37 °C with 5% $CO_2$. Type II human primary alveolar epithelial (HPAE II) cells were purchased from Shanghai Zhong Qiao Xin Zhou Biotechnology (Cat No. PRI-H-00012) and cultured in the medium provided by the manufacture (Cat No. PCM-H-029).

### Plasmids
pNL-ΔEnv-Luc, pNL-ΔEnv-GFP, pNL-Env, pcDNA3.1-VSV-G, pcDNA3.1-EBOV-GP, pcDNA3.1-EBOV-GPΔMLD, pcDNA3.1-MARV-GP, and vectors to produce EBOV trVLPs were described previously[10,11,14,15,49]. pcDNA3.1-EBOV-GPΔMLD-HA was constructed by PCR after *EcoRI/XhoI* digestion. pcDNA3.1-NPC1-3FLAG, pCAGGS-EBOV-VP40, pCAGGS-SARS2-S-FLAG, pCAGGS-SARS1-S, and pCAGGS-MERS-S were purchased from *CamateBio*. SARS2-S, SARS1-S, and MERS-S with C-terminal deletions (ΔC19

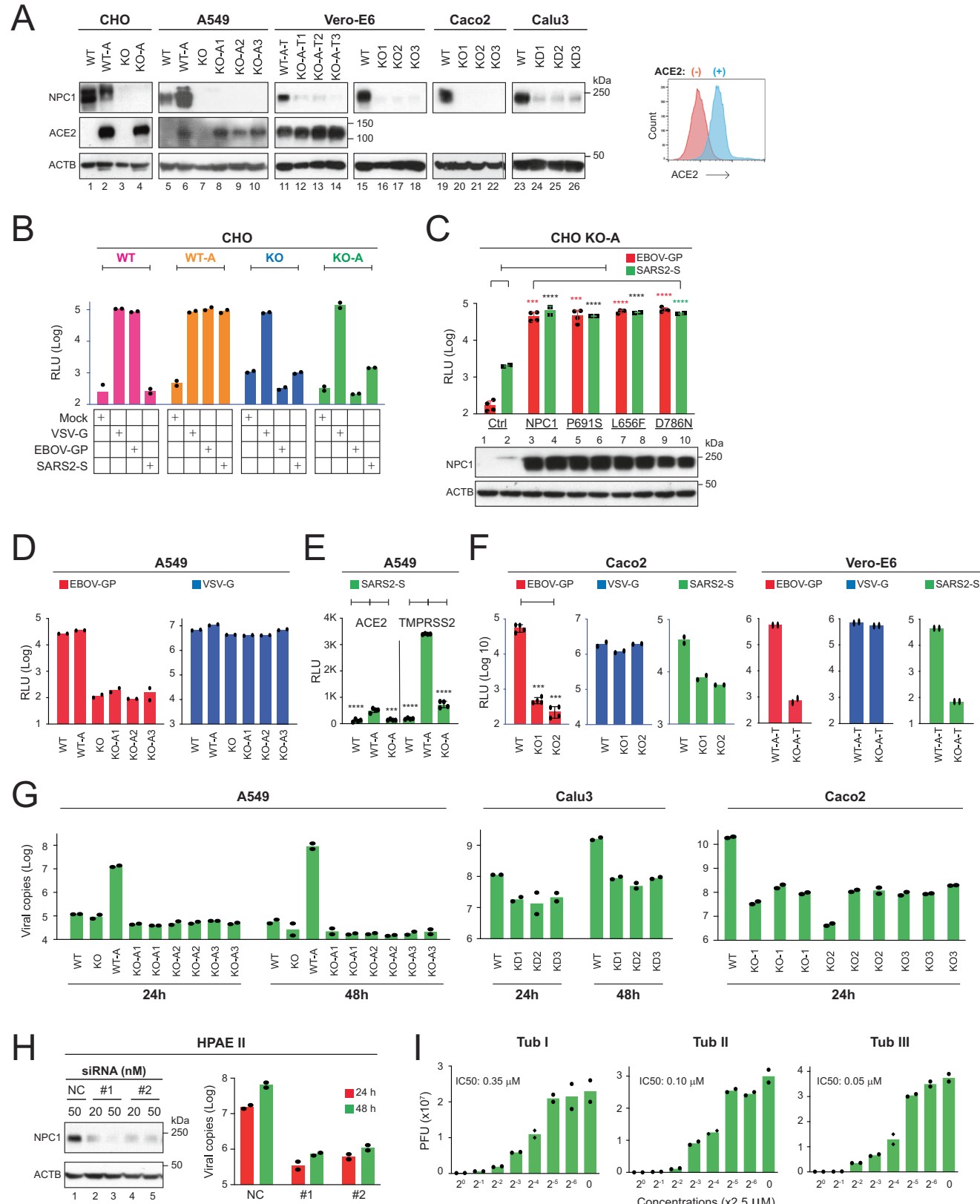

or ΔC16) and/or with N-terminal and C-terminal FLAG or HA tag were constructed in pCAGGS by PCR and *EcoR*I/*Xho*I digestion. pCAGGS-SARS2-S-Delta, pCAGGS-SARS2-S-Omicron, pCAGGS-SARS2-S-D614G, pCAGGS-SARS2-S-Alpha, pCAGGS-SARS2-S-Beta, pCAGGS-SARS2-S-Gamma, pCAGGS-SARS2-S-Kappa, and pCAGGS-SARS2-S-Lambda were provided by Sino Biological. pEGFP-N1-VP40 was constructed

by PCR after *Kpn*I digestion. pcDNA3.1-NPC1-3FLAG mutants (1-377, 1-620, L656F, P691S, D786N) were constructed by PCR followed by *Nhe*I/*Bsp*EI digestion. Human ACE2 and TMPRSS2 with a FLAG tag were expressed in pLenti-BSD after *Xba*I/*BamH*I digestion. The lentiviral CRISPR/Cas9 expression vector LentiGuide-Puro was from Feng Zhang via Addgene (#52961). The lentiviral packaging vectors pMD2.G and

**Fig. 4 | NPC1 is a critical factor for SARS2 infection. A** *NPC1* was knocked out in indicated cells (see Supplementary Figs. S5–S8). At least one KO clone was obtained from each cell line. The expression of NPC1 and ACE2 in these cells was determined by WB. The ACE2 expression on the cell surface was also determined in A549 WT (−) and WT-A (+) cells by flow cytometry after cells were gated by higher forward scatter (FSC) and side scatter (SSC). Experiments were repeated 3 times independently, and representative results are shown. **B** Indicated CHO WT and *NPC1*-KO cells were infected with HIV-1 Luc-pseudoviruses expressing VSV-G, EBOV-GP, or SARS2-S, and viral infection was determined. **C** Human NPC1 and its three mutants P691S, L656F, and D786N were expressed in CHO *NPC1*-KO cells expressing ACE2 (CHO KO-A). Cells were infected with HIV-1 Luc-pseudoviruses expressing EBOV-GP or SARS2-S, and viral infection was determined. **D** Indicated A549 cells were infected with HIV-1 Luc-pseudoviruses expressing EBOV-GP or VSV-G, and viral infection was determined. **E** Indicated A549 cells were transfected with an ACE2- or a TMPRSS2-expression vector and infected with HIV-1 Luc-pseudovirus expressing SARS2-S. Viral infection was determined. **F** Indicated Caco2 and Vero-E6 cells were infected with HIV-1 Luc-pseudoviruses expressing EBOV-GP, VSV-G, or SARS2-S, and viral infection was determined. **G** Indicated cell lines were infected with SARS2 authentic viruses. After 24 or 48 h, viral RNA copy numbers were determined by real-time PCR. **H** HPAE II cells were transfected with indicated siRNAs and infected with SARS2 authentic viruses. NPC1 expression in these cells was detected by WB and viral infection was detected by real-time PCR after 24 or 48 h of infection. **I** The $IC_{50}$ of Tubs I/II/III for authentic SARS2 were measured in Vero-E6 cells. Viral titers were determined by plaque assay after 24 h of infection. PFU, plaque forming units. Error bars in **C**, **E**, and **F** indicate SEMs ($n = 4$ biologically independent experiments). One-way ANOVA was applied. ***$P < 0.001$, ****$P < 0.0001$.

psPAX2 were from Didier Trono via Addgene (#12259, #12260). Detailed information for the construction of vectors is available upon request. Primers used for cloning are listed in Supplementary Table 2.

### HIV-1 pesudovirion production
HEK293T cells were cultured in 10-cm dish and transfected with 10 μg HIV-1 proviral vector (pNL-ΔEnv-Luc, pNL-ΔEnv-GFP) and 2.5 μg viral glycoprotein expression vector (HIV-1 Env, EBOV-GP, SARS1-S, SARS2-S, MERS-S, or VSV-G) using Polyethyleneimine (PEI). Supernatants were harvested 48 h post-transfection and centrifuged two times for 10 min at $3000 \times g$ to clear cell debris. They were further cleared by passing through a 0.22 μm filter and stored at −80 °C for infection experiments.

### High-throughput screening
Selected plant-sourced compound Library (96-well) (L4600-Selected plant-sourced compound Library) containing 974 compounds was purchased from TargetMol. Vero-E6 cells were used for infection with HIV-1 Luc-pseudoviruses expressing EBOV-GP or VSV-G and TZM-bl cells were used for infection with those expressing HIV-1 Env. These cells were cultured in 96-well plates at 40,000 cells per well in 100 μL culture medium overnight at 37 °C, 5% $CO_2$. 50 μL prediluted compounds were added into each well at 10 μM, and after 1 h, 50 μL viruses were added to cells. After 48 h, viral infection was detected by measuring intracellular luciferase activity using the Bright-Glo® Luciferase Assay System (Promega, Madison, WI, USA). Relative luciferase unit (RLU) activity was measured using Enspire® Multimode reader (PerkinElmer, Waltham, MA, USA).

Compared to DMSO control, an average luciferase signal for EBOV infection was $7.11E + 06 \pm 1.52E + 05$ and for HIV-1 was $4.84E + 06 \pm 4.67E + 05$ relative luciferase unit (RLU), which indicates a signal to background ratio of $>10^3$, and a calculated window coefficient (Z' factor) of $0.5 \pm 0.02$. The luciferase signal standard error was ±50%, and >90% inhibition of luciferase activity at 10 μM concentration was set the condition/criterion for labeling a compound as a "hit". Initial designated "hit" compounds were further tested for $IC_{50}$ values on HIV-1 pseudovirions expressing EBOV-GP and S proteins from SARS1, SARS2, and MERS. The cytotoxicity of "hit" compound was determined by ATP production assay (see below). Hits were further confirmed by dose-response response.

### Cytotoxicity assay
CellTiter-Glo® Luminescent Cell Viability Assay kit (Promega) was used to check the $CC_{50}$ of Tubeimosides. Vero-E6 cells were seeded at 35,000–40,000 cells per well in 96-well black plate with transparent bottom and treated with 2-fold serially diluted Tubeimosides for 24 h. A total of 100 μL substrates were added into each well containing 100 μL DMEM culture medium and incubated for 20–30 min at room temperature. Luminescence signals were detected by GloMax 96 Microplate Reader (Promega).

### EBOV trVLP infection
EBOV trVLP producer cells (p0) were generated under BSL-2 conditions as previously described[14,15,50]. Briefly, HEK293T cells were cultured in 6-well plates ($4 \times 10^5$ cells per well) and transfected with 125 ng pCAGGS-NP, 125 ng pCAGGS-VP35, 75 ng pCAGGS-VP30, 1 μg pCAGGS-L, 250 ng p4cis-VRNA-Rluc, 250 ng pCAGGS-T7 using PEI. After 48 h, supernatants were collected from producer cells (p0) followed by centrifugation at 3000 x $g$ for 10 min and stored at −80 °C. To generate EBOV trVLP target cells (p1), HEK293T cells were cultured in 96-well plates at 20,000 cells per well and transfected with 10.4 ng pCAGGS-NP, 10.4 ng pCAGGS-VP35, 5.88 ng pCAGGS-VP30, 81.6 ng pCAGGS-L, and 20.2 ng pCAGGS-Tim1 using PEI. After 24 h, the media was exchanged for 100 μL growth medium and cultured for an additional 2-3 h to adjust the cell condition. After replacement with 50 μL medium containing different concentrations of compounds for 1 h, 50 μL viruses harvested from producer cells (p0) were added into each well, and cells were cultured for additional 72 h. Cells were lysed in 50 μL lysis buffer, and viral infection was detected by Renilla-Glo® Luciferase Assay System (Promega) using Enspire® Multimode reader (PerkinElmer).

### Authentic SARS-CoV-2 infection
Infection with authentic SARS-CoV-2 was performed in the biosafety level 3 facility in Harbin Veterinary Research Institute of Chinese Academy of Agricultural Science as we reported previously[23]. Briefly, cells were infected with SARS-CoV-2/HRB25/human/2020/CHN (HRB25, GISAID access no. EPI_ISL_467430) at a Multiplicity of Infection (MOI) of 0.01 for 1 h. Cells were then washed three times with cold PBS containing 2% FBS and continued to be cultured for 24 h. Viral titers were determined by real-time quantitative RT-PCR to measure the viral genomic RNA copy numbers. Briefly, 140 μL supernatant was collected from each infection to extract viral RNAs using a QIAamp vRNA Minikit (Qiagen, Hilden, Germany). Viral cDNAs were synthesized using HiScript II Q RT SuperMix (Vazyme, Nanjing, China), and viral RNA copy numbers were quantified by PCR using primers targeting the viral *N* gene, which are listed in Supplementary Table 3. A vector expressing the full-length *N* gene (pBluescriptIISK-N, 4221 bp) was used as a template to generate a standard curve and calculate the copy numbers.

Viral titers were also determined by Plaque Forming units (PFUs) assay. Vero-E6 cells were inoculated with serially diluted supernatants for 1 h at 37 °C. Cells were then washed three times and plaque media was overlaid onto the cell monolayers. After being cultured for another 48 h, cell monolayers were stained with crystal violet and plaque numbers were counted.

### Influenza virus infection
A549 cells ($1 \times 10^6$/well) were seeded into 6-well plates and treated by 0.5 μM Tubeimosides 1 h before infection. Cells were then infected with A/WSN/1933 (H1N1) at an MOI of 0.1. After being washed two times with cold PBS, cells were cultured for 24 h in DMEM containing 5% (vol/

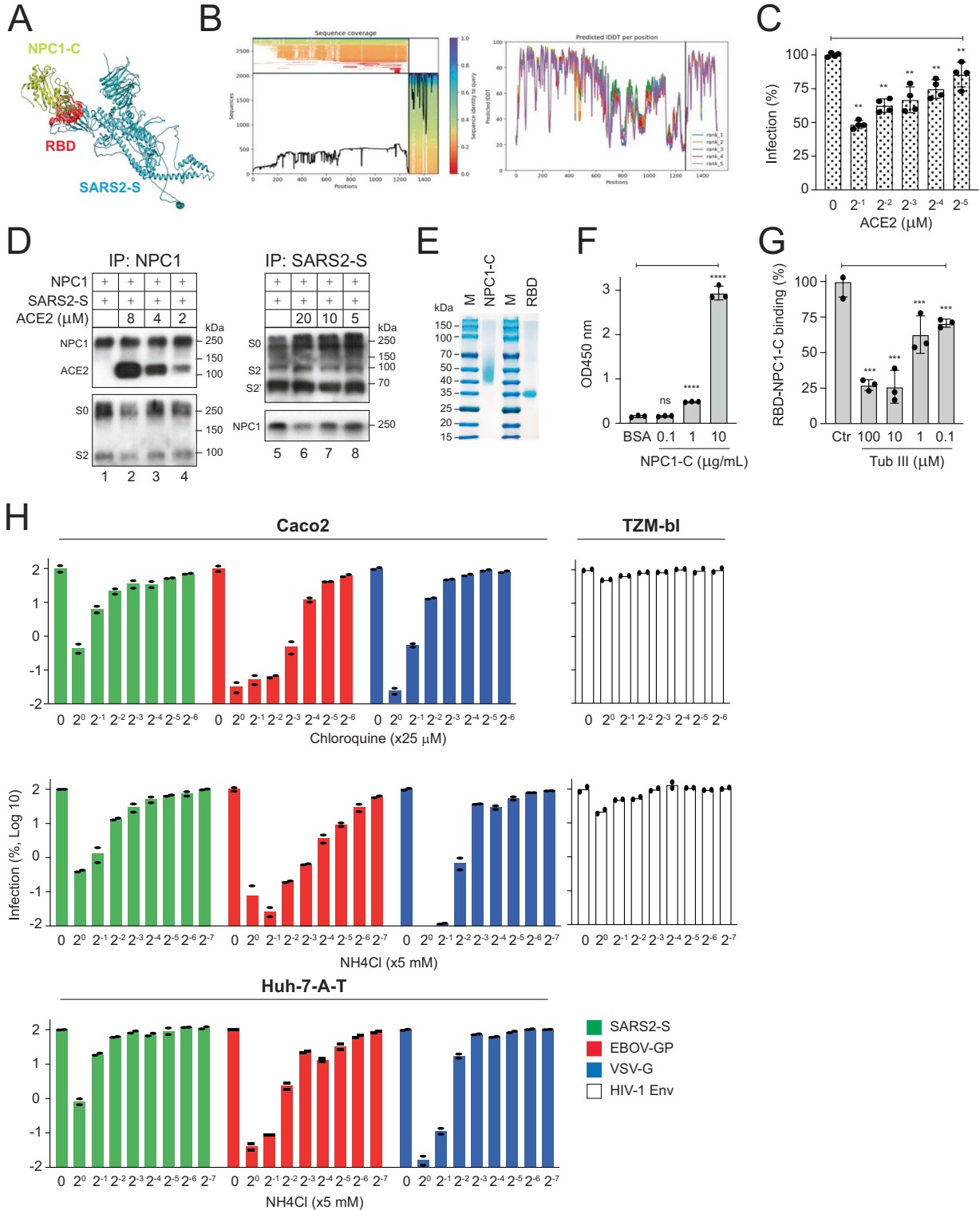

vol) fetal bovine serum (FBS) and 0.5 ug/ml TPCK in the presence of the compounds above. Cells were then collected and lysed with Trizol reagent to extract total RNAs. Cellular RNAs were reverse-transcribed with an RT Kit (FSQ-101, TOYOBO), and viral nucleoprotein (NP) and cellular glyceraldehyde-3-phosphate dehydrogenase (GAPDH) genes were quantified by a real-time quantitative RT-PCR (QPK-201, TOYOBO). The primers used are listed in Supplemental Table 3. In addition, A549 WT and *NPC1*-knockout cells were also infected, and viral infection was measured similarly, in the absence of treatment with these compounds.

**EBOV internalization**

EBOV VLPs were produced from HEK293T cells in 10-cm dish by transfection with 6 μg eGFP-VP40 expression vector and 9 μg EBOV-GP or VSV-G expression vector using PEI. Supernatants were harvested twice at 24 h and 48 h post-transfection and combined supernatants were centrifuged 10 min at 3,000 × g at 4 °C to clear cell debris. VLPs were pelleted down by ultracentrifugation through a 20% sucrose cushion at 100,000 × g for 3 h at 4 °C using Beckman Coulter J-26XPI (JCN17G24) rotor and resuspended in pre-cold PBS overnight. Vero-E6 cells cultured in 12-well plate were treated with various compounds for

**Fig. 5 | NPC1-C binds SARS2 RBD. A** Predicted NPC1-C and SARS2-S complex. The SARS2-S structure is shown in cyan, while the NPC1-C is depicted in lime yellow. The binding interface of SARS2-S in RBD, predicted to directly interact with NPC1-C, is highlighted in red. **B** NPC1-C (1274-1517 residues) and SARS2-S (1-1273 residues) complex sequence coverage and pLDDT scores are shown. **C** CHO WT-A cells were incubated with recombinant soluble ACE2 at indicated concentrations for 1 h and infected with HIV-1 Luc-pseudovirus expressing SARS2-S. After 48 h, viral infection was determined. Results are shown as relative values, with the infection in the absence of ACE2 set as 100. **D** HIV-1 pseudovirions expressing SARS2-S were cleaved by TL and incubated with purified recombinant NPC1 in the presence of increasing amounts of recombinant ACE2. Proteins were pulled down by anti-FLAG (NPC1) or anti-HA (SARS2-S) and analyzed by WB. **E** Purified NPC1-C and SARS2 RBD proteins were analyzed by SDS-PAGE followed by Coomassie blue staining. **F** The

binding of NPC1-C to SARS2 RBD at indicated NPC1-C concentrations were determined by ELISA and measured by optical density (OD) at 450 nm. **G** The inhibition of SARS2 RBD and NPC1-C binding by Tub III at indicated concentrations were determined by ELISA at OD450 nm. Results are shown as relative values, with treatment by DMSO (control, Ctr) set as 100. **H** Indicated cells were treated with increased concentrations of chloroquine or ammonium chloride (NH$_4$Cl) and infected with HIV-1 Luc-pseudoviruses expressing indicated fusion proteins. Levels of infection were calculated as relative values, with those from untreated cells set as 100. Results are shown at Log scales. Error bars in **C**, **F**, and **G** indicate SEMs ($n = 4$ (**C**) or $n = 3$ (**F**) (**G**) biologically independent experiments). One-way ANOVA was applied. **\*\*P < 0.01, \*\*\*P < 0.001, \*\*\*\*P < 0.0001**; n.s. not significant. Experiments in **D** and **E** were repeated 3 times independently, and representative results are shown.

1 h and spinoculated with VLPs at 300 × g for 1 h at 4 °C. Cells were cultured at 37 °C for 3 h and treated with 0.5% Trypsin-EDTA (Gibco) for 2-3 min to remove cell surface-associated VLPs. After washing one time with PBS, fresh media was added, and cells were cultured overnight at 37 °C. After that, cells were collected, fixed with 4% paraformaldehyde, and analyzed by flow cytometry (Cytomics TM FC 500).

### Virus interference
EBOV VLPs were produced from HEK293T cells after transfection with 9 μg EBOV-GP expression vector and 6 μg EBOV-VP40 expression vector in 10-cm dish. SARS2 VLPs were produced similarly by transfection with 4 μg S, 3 μg N, 4 μg M, and 1.5 μg E expression vectors. These VLPs were purified via ultracentrifugation at 100,000 × g for 3 h at 4 °C over a 20% sucrose cushion using Beckman Coulter J-26XPI (JCN17G24) rotor and resuspended in PBS. Huh-7 cells for EBOV infection and Huh-7-A-T cells for SARS2 infection seeded in 96 well plates were spinoculated with SARS2 or EBOV VLPs, respectively, for 1 h at 4 °C using 300 × g. Cells were cultured in at 37 °C incubator for 3-4 h and infected with pseudovirions for 48 h. Viral infection was detected by measuring intracellular luciferase activity.

### Generation of ACE2 and TMPRSS2 stable expression cell lines
Lentiviruses were produced from HEK293T cells in 10-cm dishes by co-transfection of 10 μg pLenti-ACE2-BSD or pLenti-TMPRSS2-BSD, 5 μg pcDNA3.1-VSV-G-FLAG, 6 μg psPAX2. CHO, A549, and Vero-E6 cell lines were infected with these viruses and stable cell lines were selected by treatment with blasticidin (10 μM).

### Generation of *NPC1*-knockout (KO) cell lines
Oligos encode small guide (sg) RNAs that target *NPC1* genes in human cells, Vero-E6 cells, and CHO cells (listed in Supplementary Table 4) were cloned into LentiGuide-Puro vector after *BsmB*I digestion. Lentiviruses were produced from HEK293T cells by transfection of Lenti-Guide-Puro, pMD2.G, and psPAX2 at a ratio of 4:3:2 using PEI. After infecting cells for 48 h, cells were selected by treatment with puromycin at 10 μM for CHO cells, 5 μM for A549 cells, 6 μM for Caco2 cells, 10 μM for Vero-E6 cell, and 2 μM for Calu3 cells. Single clones of CHO, A549, Caco2 and Vero-E6 were sorted in 96-well plates using Beckman Coulter MoFlo XDP cell sorter. Calu3 was used directly after treatment by puromycin. *NPC1*-KO cells were identified by western blotting.

### *NPC1*-knock down by siRNAs
HPAE II cells were transfected with *NPC1* siRNAs or a negative control siRNA (Supplementary Table 4) using INTERFERin transfection reagent from Polyplus (Cat No. 409-10). Cells were then infected with authentic SARS-CoV-2 and viral titers were determined by real-time PCR.

### Cathepsin B and L enzymatic activity
A549 cells were seeded in a 96-well plate and treated with Tubeimosides for 24 h. Cells were lysed with M2 lysis buffer (25 mM Tris-

Phosphate (pH 7.8), 2 mM 1,2-diaminocyclohexanase-N,N,N,N-tetra acetic acid, 2 mM DL-Dithiothreitol (DTT), 10% glycerol, 1% Triton X-100) for 20–30 min at room temperature, followed by adding CTS-B and CTS-L substrates at 5 μM. After 30 min incubation at room temperature, luminescence signals were detected by Enspire® Multimode reader (PerkinElmer). CTS-B and CTS-L activity were detected by substrate Bz-Arg-4MβNA, or Z-FR-AMC, respectively.

### Endosomal acidification
A549 cells were seeded in 24-well plates and treated with 0.5 μM Tubeimosides or 100 mM ammonium chloride (NH$_4$Cl) for 4–6 h. Cells were then incubated with Lyso-Tracker Red (Molecular Prob) (1:20000 dilution) for 60 min and visualized by fluorescence microscope (EVOS FL Auto Imaging System).

### Detection of intracellular cholesterol accumulation
SNB-19 cells were cultured on glass coverslip in a confocal dish and treated with various inhibitors. After being fixed with 4% paraformaldehyde, cells were washed twice with PBS, and incubated with 50 μg/ml filipin III in PBS for 1 h in dark at room temperature. After being washed 3 times with PBS, they were visualized by Zeiss Axio Observer fluorescence microscope (LSM980) using the DAPI channel.

### Detection of NPC1 binding to viral proteins on virions
To purify NPC1 proteins, HEK293T cells were cultured in 13 10-cm dishes and transfected with 9 μg pcDNA3.1-NPC1-3xFLAG per dish using PEI. After 48 h, cells were lysed with 1 mL pre-cold buffer (50 mM Tris, pH 7.5, 150 mM NaCl, 5% glycerol, 1% DDM) with anti-protease inhibitor cocktail per dish at 4 °C for 30–40 min. The whole cell lysate was further centrifuged at 13,000 × g for 10 min at 4 °C to get rid of insoluble fractions. Supernatants were mixed with anti-FLAG beads (Sigma) and rotated overnight at 4 °C. These NPC1-captured beads were collected and washed 5 times with PBS. HIV-1 pseudovirions were produced from HEK293T cells by transfection with pNL-ΔEnv-Luc and a vector expressing EBOV-GP or SARS2-S. Virions were purified by ultracentrifugation at 100,000 × g for 3 h and resuspended in PBS. Purified virions were digested with thermolysin for 1 h at 37 °C and digestion was terminated by adding 10 mM Phosporamidon. Digested virions were incubated with NPC1-captured beads for 1 h at room temperature. Unbound virions were removed by washing beads with PBS 4-5 times. Bead-associated proteins were detached by adding the lysis buffer along with 1x SDS-PAGE loading buffer and analyzed by Western blotting.

### Immunoprecipitation
HEK293T cells were cultured in 60-mm dish and transfected with 3 μg NPC1 expression vectors (full-length, 1-377, 1-620) and 4 μg viral protein expression vector (EBOV-GP, SARS1-S, SARS2-S, MERS-S, or VSV-G) using PEI. After 24–36 h, cells were lysed with the same lysis buffer for detection of the NPC1 binding to SARS2-S protein for 30–40 min at 4 °C. The whole cell lysate was collected in 1.5 mL tubes and

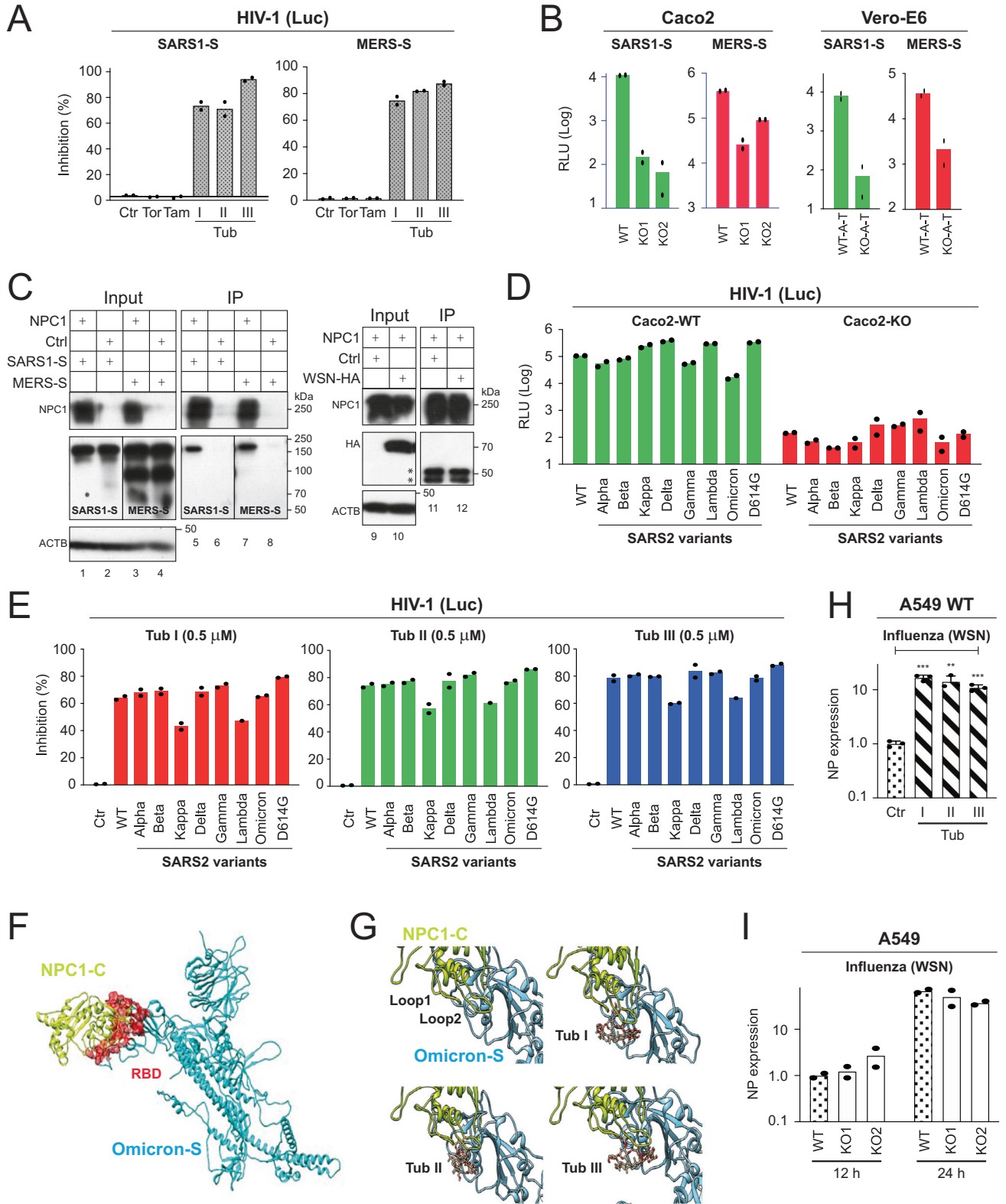

centrifuged at $13,000 \times g$ for 10 min at 4 °C. Supernatants were collected and proteins were pulled down with anti-FLAG beads (Sigma). The interaction of NPC1 with viral proteins were determined by Western blotting.

## Detection of cell surface ACE2
A549 and A549-ACE2 cells were stained with the primary anti-ACE2 antibody (R&D, cat: AF933) and then secondary APC-conjugated anti-

goat antibody (Jackson, Cat No. 705-136-147). After being washed twice with PBS, stained cells were analyzed by flow cytometry using Beckman Cytomics TM FC 500. Results were analyzed with FlowJo.

## Western blotting (WB)
Cells were lysed in ice-cold RIPA lysis buffer (25 mM Tris, pH 7.4, 150 mM NaCl, 0.5% sodium deoxycholate, 0.1% SDS, 1% Nonidet

**Fig. 6 | NPC1 is a critical factor for SARS1, MERS, and SARS2 VOC infection.**
**A** Huh-7-A-T cells were treated with indicated compounds at 1 μM and infected with
HIV-1 Luc-pseudoviruses expressing SARS1-S or MERS-S and viral inhibition was
determined as previously. DMSO was used as a control (Ctr). **B** Caco2, Vero-E6, and
their *NPC1*-KO cells were infected with HIV-1 Luc-pseudoviruses expressing SARS1-S
or MERS-S, and viral infection was determined. **C** NPC1 was expressed with SARS1-S,
MERS-S, and influenza virus A/WSN/1933 (H1N1) HA in HEK293T cells. After
immunoprecipitation (IP), proteins in cell lysate (input) and pulldown samples were
analyzed by WB. (*) indicates non-specific bands. **D** Caco2 WT and *NPC1*-KO cells
were infected with HIV-1 Luc-pseudoviruses expressing indicated S proteins and
viral infection was determined. **E** Huh-7-A-T cells were treated with Tubs I/II/III at
0.5 μM and infected with HIV-1 Luc-pseudovirions expressing indicated S proteins
from SARS variants. Viral inhibition was determined as previously. DMSO was used

as a control (Ctr). **F** Predicted NPC1-C and Omicron-S complex. The Omicron-S
structure is shown in cyan, while the NPC1-C is depicted in lime yellow. The binding
interface of Omicron-S in RBD, predicted to directly bind NPC1-C, is highlighted in
red. **G** Docking of Tubs I/II/III to the interphase of NPC1-C and Omicron-S protein
complex was obtained via Webina. **H** A549 cells were treated with 0.5 μM Tubs I/II/
III for 1 h and infected with influenza virus A/WSN/1933 (H1N1). After 24 h, viral
nucleoprotein (NP) genes were quantified by RT-PCR. Results are shown as relative
values, with those from cells treated with DMSO set as 1. DMSO was used as a
control (Ctr). Error bars indicate SEMs ($n = 3$ biologically independent experi-
ments). one-way ANOVA was applied. **$P < 0.01$, ***$P < 0.001$. **I** A549 WT and *NPC1*-
KO clones were infected with H1N1, and NP genes were quantified the same as in
**H** at indicated time points.

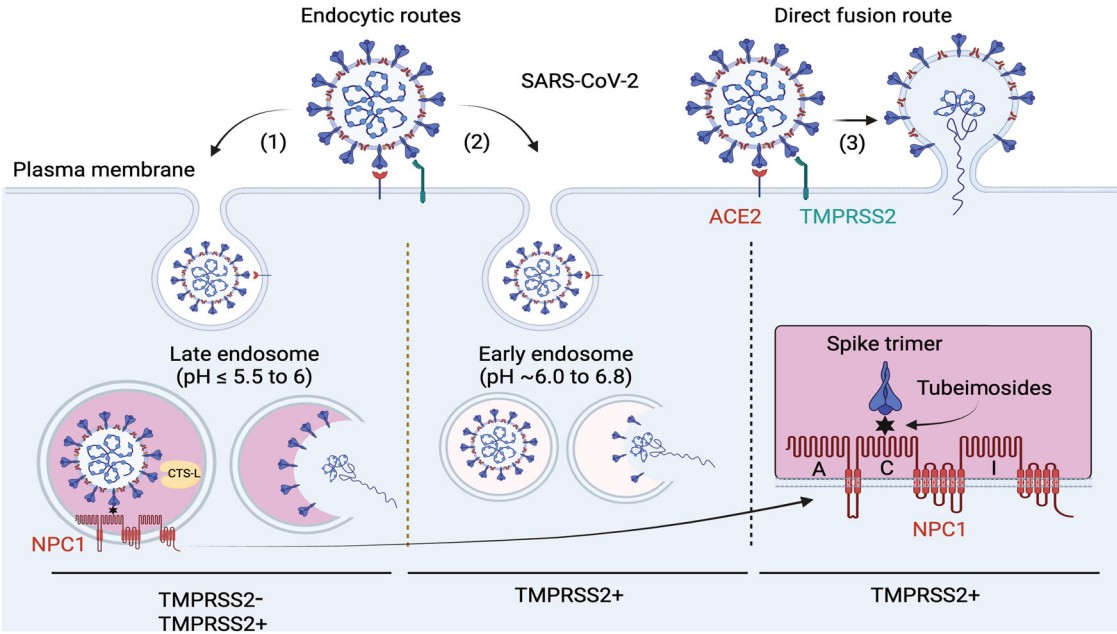

**Fig. 7 | The role of NPC1 in SARS2 entry.** SARS2 enters cells via late endosomes (1),
early endosomes (2), and the plasma membrane (3). The late-endosome entry
occurs in TMPRSS2⁻ cells at neutral pH and TMPRSS2⁺ cells at mildly acidic pH.
During this process, S proteins are either cleaved by CTS-L in late endosomes or by
TMPRSS2 on the cell surface. Processed S proteins bind NPC1-C via RBD to com-
plete membrane fusion and cell entry, which is blocked by Tubeimosides. This
figure was created by BioRender (https://www.biorender.com).

P-40 [Sigma-Aldrich, R0278]) supplemented with protease inhi-
bitors cocktail (Sigma-Aldrich, P8340). Approximately 0.1 mL
RIPA was used for a total of $2 \times 10^6$ cells. Cell lysate was cen-
trifuged $12,000 \times g$ at 4 °C for 10 min and supernatants were col-
lected. Samples were boiled in SDS-polyacrylamide gel
electrophoresis (SDS-PAGE) loading buffer (Solarbio Life Sci-
ences; P1015) and resolved by SDS-PAGE. Separated proteins were
transferred onto PVDF membranes and blocked with 5% nonfat
milk powder in TBST (Tris-buffered saline [20 mM Tris, pH 7.4,
150 mM NaCl, 0.1% Tween 20]); Solarbio Life Sciences, T8220) for
1 h at room temperature. EBOV-GP was detected with rabbit anti-
EBOV-GP (Sino Biology, China) antibodies (1:5000 dilution); S2
proteins of SARS2, SARS1 and MERS were detected with rabbit
anti-SARS CoV-S antibodies (Novus Biologicals, NB100-56,047)
(1:10,000 dilution); ACE2 was detected with goat anti-hACE2
(1:5000); NPC1 was detected with rabbit anti-NPC1 (Abcam)
(1:5000 dilution). Membranes were incubated with these primary
antibodies overnight at 4 °C and washed 4 times with 1x TBST
buffer at a duration of 20 min each time. HRP-conjugated sec-
ondary antibodies were added at 1:5000 dilution for 1 h at room
temperature, followed by washing with 1x TBST buffer. FLAG-tag,
HA-tag, and actin were detected with anti-FLAG-HRP, anti-HA-
HRP, and anti-Actin-HRP, respectively at a dilution 1:10,000

dilution. HRP signals were detected with ECL-Supersensitive
Luminescence Liquid (Beijing Applygen Technologies; P1010),
followed by exposure to X-Ray film.

### Prediction of protein-protein binding interface
To define the protein-protein binding interface in the predicted com-
plexes, we employ alphashape analysis[51]. All predicted WT-S/NPC1-C
and Omicron/NPC1-C complexes were analyzed, and additional details
can be found in Fig. S10 and S13, and Table S1.

### ELISA
To detect the SARS2-S RBD and NPC1-C binding, an ELIAS plate was
coated with an anti-RBD at 1:200 at 4 °C overnight. The plate was
blocked with 1x ELISA/ELISAPOT Diluent Assay buffer (Cat No. 88-
7066) from ThermoFisher Scientific for 1 h at room temperature and
then incubated with recombinant SARS2 RBD at 0.5 mg/mL. After
washing with PBS with 1% Tween-20 (PBS-T) three times, recombinant
NPC1-C protein was added at 0.1 μg/mL, 1 μg/mL, and 10 μg/mL in 1x
ELISA/ELISAPOT Diluent Assay buffer. After 5-h incubation followed by
washing with PBS-T 3 times, an HRP-conjugated anti-FLAG was added
at 1:400 dilution and incubated at room temperature for 1 h. After
further washing, 3,3',5,5'-Tetramethylbenzidine (TMB) was used as the
substrate to detect the HPR signal, which was measured by a

microplate reader at 450 nM after the reaction was stopped by 2 M $H_2SO_4$. To detect the Tub III activity, NPC1-C was used at 1 μg/mL and Tub III was used at 100 μM, 10 μM, 1 μM and 0.1 μM.

## Graphic preparation

Figures were prepared using Adobe Illustrator 2021. The model in Fig. 7 was created by BioRender (https://biorender.com).

## Statistical analysis

All experiments were performed independently at least three times, and statistically analyzed by Student's *t* test using GraphPad prism 9. Quantitative values of data were expressed as mean ± standard error of measurements (SEMs) and represented by error bars. Comparisons were analyzed by one-way analysis of variance (ANOVA) followed by Tukey test. A *p* value < 0.05 ($p < 0.05$) was statistically significant when *$p < 0.05$, **$p < 0.01$, ***$p < 0.001$, ****$p < 0.0001$, ns (not significant, $p > 0.05$).

## Reporting summary

Further information on research design is available in the Nature Portfolio Reporting Summary linked to this article.

## Data availability

Source data are provided with this paper.

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

## Acknowledgements

S.L. is supported by a grant from the National Natural Science Foundation of China (32172836). X.P.W. is supported by grants from the Central Public-interest Scientific Institution Basal Research Fund (1610302022002) and the Natural Science Foundation of Heilongjiang Province of China (LH2022C110). Y.H.Z. is supported by a grant from the National Institutes of Health (AI164266). We thank Terry Moore for drawing compound structures (Fig. 1B). Figure 7 was created by BioRender (https://www.biorender.com).

## Author contributions

I.K., X.P.W. and S.L. conducted most of the experiments; L.T. and C.W. conducted SARS-CoV-2 infection in BSL3; G.Z., Z.W., J.W. and Z.G.B. coordinated infection experiments in BSL3; I.A., W.S. and R.H.H. contributed reagents; H.L. and A.M. conducted docking and structural analysis; B.Y. and J.L. conducted alphashape analysis; X.L. provided technical assistance; I.K., X.P.W. and Y.H.Z. designed experiments; Y.H.Z. wrote the manuscript with input from all authors.

## Competing interests

The authors declare no competing interests.
