## [Peer Review File · Nature Communications]

Tubeimosides are pan-coronavirus and filovirus inhibitors that can block their fusion protein binding to Niemann-Pick C1REVIEWER COMMENTS

Reviewer #1 (Remarks to the Author):

Previous work has demonstrated that NPC1, the endosomal receptor for Ebola and other filoviruses, is needed for the entry of SARS-CoV-2 (SARS2)(Ref. 20). Here the authors propose that this is due to a direct interaction between the receptor binding domain (RBD) of the SARS2 spike protein (SARS2S) and NPC1. This interaction was modeled in silico (Fig. 5A), but to this reader its experimental test (Fig. 5D) does not provide sufficiently compelling evidence (Major Comment 1). Hence, while the authors have (i) confirmed a role for NPC1 for SARS2 entry (and extended this need to other CoVs), additional work is needed to support a direct interaction between the SARS2-S RBD and NPC1 (Fig. 7) and its inhibition by tubeimosides.

Main Comments

1. RE: the proposed interaction between SARS2S-RBD and NPC1: Beyond the in silico work (Fig. 5A), the SARS2S-RBD--NPC1 interaction was inferred based on interference by recombinant ACE2 (Fig. 5D), which was argued as meaningful based on ACE2's known binding to the RBD. But (a) this is an indirect test, (b) the NPC1 employed may not have been in a native state (see Minor Comment 11), and (c) the observed inhibitions are modest. Hence the interaction between SARS2-S-RBD and NPC1 needs to be scrutinized with additional tests. Short of a structural demonstration of the interaction (e.g., by x-ray or cryo EM), more precise specificity tests are needed, e.g., an ELISA as in Ref. 8 with purified soluble NPC1 C-loop and purified recombinant SARS2 RBD, bolstered with point mutations in the predicted SARS2S-RBD- NPC1 interface

2. Another feature of the summary model (Fig. 7) requires experimental bolstering or, minimally, questioning in the Discussion: that even in TMPRSS2-expressing cells, SARS2 particles must traffic to NPC1+ late endosomes and experience late endosomal low pH for entry. The authors of the manuscript under consideration support this contention based on Refs. 4 and 21, which discuss a low pH requirement for SARS2 entry, stating (L301) that there is "agreement" on this point. But, Ref. 21 reports two mutually exclusive pathways: TMPRSS2-dependent, low pH-independent through the cell surface and TMPRSS2-independent, low pH dependent through endosomes; the two pathways can reside in the same cells, with relative amounts of TMPRSS2 dictating the degree to which each pathway is used in different cell types. Also, the single report for low pH dependent TMPRSS2-dependent entry (Ref. 4) showed a need for only mildly low pH (~pH 6.7) for TMPRSS2-dependent entry, which is more consistent with entry through early vs. NPC1-containing late endosomes. Moreover, many studies have demonstrated SARS2-mediated cell-cell fusion with TMPRSS2-expressing cells at neutral pH*, and cited work (Ref. 35) using a biophysical SARS2S particle fusion assay has demonstrated fusion at neutral pH. [*From the Methods section and legend, it appears that the cell-cell fusion data in this manuscript (Fig. 4D) were acquired in neutral pH medium.] Hence the exact pH requirements for SARS2S fusion and its exact site of entry in TMPRSS2+ cells are still being clarified.

Minor Comments

1. Lines 41-42: Edit to inform that paxlovid is FDA approved and molnupiravir is approved under an Emergency Use Authorization (EUA).
2. Line 58: edit: 'reported' vs. 'shown'. (The acid pH requirement has only been shown in one study (Ref. 4), and other studies support pH-independent fusion for SARS2
3. Ref. 20, which showed that U18666a and KO of NPC1 inhibit SARS2 infection, should be mentioned in the Introduction. For example it would be appropriate for Line 70 to read: "and confirmed findings (Ref. 20) indicating that entry is dependent on NPC1"
4. Line 110: edit: "which may explain their less potent..."
5. Fig. 2B: edit: 'Cholesterol accumulation' (vs. 'cholesterol trafficking')
6. Line 119: edit: 'by interfering with the function of NPC1' (vs. 'by targeting NPC1')
7. To further assess the mechanism of tubeimosides vs. Ebola, the authors should test if tubeimosides block arrival of EBOV-GP particles to late endosomes (co-localization w/late endosome markers) and if they block at the level of fusion using viral-like particles (VLPs) tagged

with beta-lactamase (e.g., as in Ref. 12).

8. Line 125: edit: 'EBOV-GP pseudovirus infection' (vs. EBOV infection)'

9. Line 148: 'Tam' (vs. 'Tub')?

10. Fig. 3E-H legend: specify whether (as per Methods) RIPA or M2 lysis buffer was used to prepare the cell lysates. Standard RIPA buffers will denature many proteins, and so it would be preferable to prepare cell lysates for all of these experiments in a non-denaturing cell lysis buffer (e.g., the M2 lysis buffer, which the reader presumes is non-denaturing).

11. RE: Figs. 3H and 5D: As above, what detergent was used to lyse the cells to purify the NPC1 protein? If RIPA, NPC1 may have been denatured during the purification. Also, whatever detergent was used for the purification, being a 13-pass transmembrane protein, NPC1 is likely not in a native state on the beads used for these experiments, which were washed 5-times with PBS.

12. RE: Fig. 4: (a) The data in Fig. 4D should be quantitated. Ideally, the authors would perform an inherently more quantitative cell-cell fusion assay (e.g., one employing split luciferase); (b) Why are the background levels in the CMFDA channel variable? (c) Most importantly: the authors should comment on why KO of NPC1, an endosomal protein, should inhibit cell-cell fusion, which is a cell surface phenomenon.

13. RE: the effects of tubeimosides on Ebola GP1 interaction with NPC1. This would be more incisively demonstrated using purified NPC1 C-loop and purified Ebola GP1 trimeric ectodomain, e.g., in an ELISA as in Ref. 8, with analogous experiments performed for the effects of tubeimosides on an interaction between SARS2S-RBD and NPC1 C-loop.

14. There is also some concern re: interactions reported based on co-IP experiments from 293T cells exogenously expressing NPC1 and viral glycoproteins due to likely over-expression of these proteins and questions about whether they maintain native states (e.g, for the SARS1 and MERS S experiments in Fig. 6C)

15. Fig. 6H: It is curious that tubeimosides and NPC1 KO increase influenza infection (Figs. 6H, I). The authors should comment on why they think this is so.

16. L302: "low" (vs. "high")?

17. L313-314: Clarify: Ref. 35 showed that ACE2 is not required for fusion when an alternate means of attaching SARS2S particles to target membranes is provided.

18. Lines 330-331: potential therapeutic utility of tubeimosides. What is known about the pharmacology of tubeimosides? Can they be given orally? By any route known, are exposure levels (e.g., Cmax and/or AUC) high enough to have an anti-viral effect. This information is needed to support therapeutic utility. If not known, the authors should delete the word 'great' on Lines 28 and 330.

Reviewer #2 (Remarks to the Author):

The manuscript entitled "SARS-CoV-2 requires Niemann-Pick C1 (NPC1) to complete cell entry" by Khan et al., presents a logical progression of experiments that cumulatively demonstrate that the host factor NPC1 is involved in SARS-CoV-2 entry. There are strengths and weaknesses to consider.

Major Comments:

1) One perceived weakness is that there are previous published studies that make the conclusion that NPC1 is a SARS-CoV-2 entry factor. That being said, the current study does provide the most extensive analysis. And further, based on this extensive analysis, including the first evidence of direct interaction of SARS-CoV-2 Spike and NPC1, the current report does propose a different mechanism/role for NPC1 than previous studies (which propose a more indirect role of NPC1 in regulating cholesterol levels, ACE2 levels, and even perhaps co-localizing with late endosomes that have more protease activity)(refs 1-4 below). Thus, the current study is novel in the mechanistic conclusions drawn. Specific suggestions:

a. It is important for the authors to highlight the different mechanism proposed by others and discuss how their data fits into the pre-existing literature.

b. It would be particularly compelling for the authors to test in their hands and in their

experimental models, the mechanisms proposed by others (e.g., does adding back exogenous cholesterol reverse the effects on viral entry as reported, ref.1)

2) The conclusion that TMPRSS2-dependent entry relies on late endosomes (Line 245, Fig. 7, and discussion) is not supported.

a. The data in Fig. 4F could be explained the other indirect mechanisms proposed for NPC1 involvement in SARS CoV-2 entry which is unrelated to endocytosis or late endosomes, but rather changes in cholesterol at the plasma membrane and/or reduction in ACE2 levels. Considering previous reports, it is important to consider these alternative explanations before suggesting the well documented TMPRSS2 pathway has an endocytic component.

b. The data in Fig. 5E shows that the effect of acidification inhibitors has less effect on SAR-CoV-2 than it does on Ebola and VSV-G, which is consistent with one SARS CoV 2 entry pathway independent of endocytosis.

3) Throughout, key information and statistical analysis is missing from the data/figures.

a. No analysis is included indicating which effects are statistically significant (i.e., p values)

b. While often the unique data points for experiments are shown, this is not always the case, and the dots are not defined (e.g., are these technical replicates or biological replicates?). It needs to be clear how many biological replicates were included in each experiment, how many times each experiment was repeated, and what the apparent "error bars" represent. For example:

- Fig. 1C - It looks like there are two types of "error bars", what are each of these showing?

- Fig. 1E - What does the line represent? Is this the average of the data points shown? Are these data points biological replicates or from 3 different experiments?

Minor Comments:

1) What are the 19 compounds that were considered to be hits in Fig. S2? It is not clear why Tub1 would have been chosen. Please mark those compounds considered hits. ("decreased" mis-spelled on the y-axis)

2) Fig. S4, there appears to be a dose-dependent inhibition of VSV (although perhaps p values would indicate the differences are not significant. Perhaps this is related to the alternative mechanism proposed for NPC1 involving cholesterol dysregulation. If the inhibition is significant, can the authors please acknowledge this and comment on this? (Can this be inhibition be countered by addition of cholesterol?)

3) Some things are overstated and could/should be tempered.

a. Line 38 – the CoVID pandemic has not caused "unprecedented" damages. Most previous pandemics cause much more death when the population was in fact much smaller

b. Line 40 – the emergency of SARS CoV-2 VOC is not "frequent." It is relatively slow compared to other RNA viruses

References:

1. Guoli Li # 1 2, Bingqian Su # 1 2, Pengfei Fu 1 2, Yilin Bai 3, Guangxu Ding 1 2, Dahua Li 1 2, Jiang Wang 1 2 4, Guoyu Yang 5 6 7, Beibei Chu 8 9 10 . NPC1-regulated dynamic of clathrin-coated pits is essential for viral entry. *Sci China Life Sci.* 2022 Feb;65(2):341-361. PMID: 34047913.

2. Cecilia Vial 1, Juan Francisco Calderón 1, Andrés D Klein 1 . NPC1 as a Modulator of Disease Severity and Viral Entry of SARS-CoV- 2. *Curr Mol Med.* 2021;21(1):2-4. PMID: 32660402.

3. Rebecca M Mingo 1, James A Simmons 1, Charles J Shoemaker 1, Elizabeth A Nelson 1, Kathryn L Schornberg 1, Ryan S D'Souza 1, James E Casanova 1, Judith M White 2. Ebola virus and severe acute respiratory syndrome coronavirus display late cell entry kinetics: evidence that transport to NPC1+ endolysosomes is a rate-defining step. *J Virol.* 2015 89(5):2931-43. PMID: 25552710 (Report on the original SARS CoV, but relevant)

4. Rami A Ballout 1, Dmitri Sviridov 2, Michael I Bukrinsky 3, Alan T Remaley 1. The lysosome: A potential juncture between SARS-CoV-2 infectivity and Niemann-Pick disease type C, with therapeutic implications. *FASEB J.* 2020 Jun;34(6):7253-7264. PMID: 32367579 (a review which summarizes the other proposed mechanisms nicely).

Reviewer #3 (Remarks to the Author):

SARS-CoV-2 requires Niemann-Pick C1 (NPC1) 1 to complete cell entry

Previous CRISPR screens have implicated NPC1 as a host cell susceptibility factor for SARS-CoV-2 infection. This submission provides abundant additional evidence that NPC1 is used by SARS-CoV-2 to enter cells. The findings start by showing that pharmacologic inhibitors block SARS-CoV-2 and EBOV pseudovirus entry. Knowing that EBOV GP uses NPC1 for entry, the studies then follow those used previously by filovirologists in several studies of virus-NPC1 interactions. Several important findings were made. Cells lacking NPC1 are relatively resistant to pseudovirus entry SARS-CoV-2, spikes bind NPC1, binding is blocked by the inhibitors. The findings with SARS-CoV-2 pseudoviruses were extended to authentic SARS-CoV-2, which was also blocked by inhibitors and was also dependent on NPC1 for infection. The findings were also extended to SARS-CoV and to MERS-CoV, which also showed some dependence on NPC1 for entry. These are significant and important findings for the field.

Comments:

1. There are concerns about the support for statements that SARS-CoV-2 is “completely” dependent on endocytosis (line 26, lines 253-254 and elsewhere). The dependence for endosomal NPC1 would support this statement, but the results with inhibitors of endosomal acidification (Fig 5E) do not. Also, the result in Fig 4F is very interesting but cannot be used to conclude that SARS-CoV-2 entry is exclusively endosomal. The discussion on lines 295-305 might be further developed to incorporate findings on the relationships between SARS-CoV-2 spike cleavage and entry kinetics. There are several reports demonstrating that prior cleavage of SARS-CoV-2 spikes will allow for bypass of endosomal neutralizing agents and cell-surface CoV entry.
2. Extending from point 1, is it not puzzling that NPC1 facilitates cell-to-cell fusion (Fig 4D)? How is this finding reconciled with the acid pH dependence and endosomal spike-mediated fusion process that is promoted in the manuscript?
3. Discussion of the findings in conjunction with a related recent paper (<https://doi.org/10.1016/j.cell.2023.06.005>) would add impact, as this paper highlights a separate endolysosomal transmembrane protein in facilitating SARS-CoV-2 cell entry.
4. Additional consideration of the model in Fig 7 may be helpful. SARS-CoV-2 entry required ACE2, a well-known receptor that drives entry through conformational changes in spike proteins. Findings in this submission showed that SARS-2 entry was facilitated by NPC1, but only when ACE2 was also present. The discussion on lines 310-314 give the impression that a single spike trimer may bind a single ACE2 monomer, leaving two of three trimer subunits available for NPC1 binding. If this is the model, it could be made more apparent in Fig 7, which at present appears underdeveloped in relation to the findings of the paper. The model also should account for the fact that there are many SARS-CoV-2 attachment factors in addition to ACE2 that are known to bring SARS-CoV-2 into endosomes, so ACE2 is not liable to be a mere “attachment factor”, given its essential requirement for entry in all experiments of this submission.

Additional minor comments:

1. Consider leading introduction with refs 38, 39, as these findings provide a premise for focusing on NPC1.
2. Consider sticking with the historical nomenclature (SARS-CoV-2, not “SARS2”), although this may be decided by the journal and its standards for consistent nomenclature.
3. Line 48, mutations beyond S gene also produce VOC
4. Line 53, ACE2 does drive conformation changes; this correct statement appears at odds with lines 315-316, which appear to demote ACE2 to a mere “attachment factor”. Some reconsideration of the roles for ACE2 in relation to NPC1 would be helpful in the discussion.

Reviewer #1 (Remarks to the Author):

Previous work has demonstrated that NPC1, the endosomal receptor for Ebola and other filoviruses, is needed for the entry of SARS-CoV-2 (SARS2) (Ref. 20). Here the authors propose that this is due to a direct interaction between the receptor binding domain (RBD) of the SARS2 spike protein (SARS2S) and NPC1. This interaction was modeled in silico (Fig. 5A), but to this reader its experimental test (Fig. 5D) does not provide sufficiently compelling evidence (Major Comment 1). Hence, while the authors have (i) confirmed a role for NPC1 for SARS2 entry (and extended this need to other CoVs), additional work is needed to support a direct interaction between the SARS2-S RBD and NPC1 (Fig. 7) and its inhibition by tubeimosides.

We would like to thank this reviewer for these important comments. We have revised this manuscript based on these comments.

Main Comments

1. RE: the proposed interaction between SARS2S-RBD and NPC1: Beyond the in silico work (Fig. 5A), the SARS2S-RBD--NPC1 interaction was inferred based on interference by recombinant ACE2 (Fig. 5D), which was argued as meaningful based on ACE2's known binding to the RBD. But (a) this is an indirect test, (b) the NPC1 employed may not have been in a native state (see Minor Comment 11), and (c) the observed inhibitions are modest. Hence the interaction between SARS2-S-RBD and NPC1 needs to be scrutinized with additional tests. Short of a structural demonstration of the interaction (e.g., by x-ray or cryo EM), more precise specificity tests are needed, e.g., an ELISA as in Ref. 8 with purified soluble NPC1 C-loop and purified recombinant SARS2 RBD, bolstered with point mutations in the predicted SARS2S-RBD- -NPC1 interface.

As suggested by this reviewer, we have developed an ELISA to confirm the direct interaction between SARS2-S RBD and NPC1-C using purified recombinant proteins. These new results are presented in Fig.5E and Fig.5F. In addition, we also show that this interaction is blocked by Tubeimoside III in Fig.5G to validate the specificity of this interaction and the activity of Tubeimosides. As shown in Table S1, there are over 100 residues in the RBD-NPC1 interface. To map these critical residues is another big project in the lab now, so the results will be reported in our future publications.

2. Another feature of the summary model (Fig. 7) requires experimental bolstering or, minimally, questioning in the Discussion: that even in TMPRSS2-expressing cells, SARS2 particles must traffic to NPC1+ late endosomes and experience late endosomal low pH for entry. The authors of the manuscript under consideration support this contention based on Refs. 4 and 21, which discuss a low pH requirement for SARS2 entry, stating (L301) that there is “agreement” on this point. But, Ref. 21 reports two mutually exclusive pathways: TMPRSS2-dependent, low pH-independent through the cell surface and TMPRSS2-independent, low pH dependent through endosomes; the two pathways can reside in the same cells, with relative amounts of TMPRSS2 dictating the degree to which each pathway is used in different cell types. Also, the single report for low pH dependent TMPRSS2-dependent entry (Ref. 4) showed a need for only mildly low pH (~pH 6.7) for TMPRSS2-dependent entry, which is more consistent with entry through early vs. NPC1-containing late endosomes. Moreover, many studies have demonstrated SARS2-mediated cell-cell fusion with TMPRSS2-expressing cells at neutral pH*, and cited work (Ref. 35) using a biophysical SARS2S particle fusion assay has demonstrated fusion at neutral pH. [*From the Methods section and legend, it appears that the cell-cell fusion data in this manuscript (Fig. 4D) were acquired in neutral pH medium.] Hence the exact pH requirements for SARS2S fusion and its exact site of entry in TMPRSS2+ cells are still being clarified.

We greatly appreciate that this reviewer encouraged us to further discuss TMPRSS2 on SARS2 entry, which is now included in Discussion. Based on our results and those from literatures, we respectfully propose that SARS2 enters cells via late endosomes, but not the direct breakthrough of the plasma membrane by fusion. Currently, the plasma membrane entry model is only supported by indirect evidence with speculation, including the less dependency on low pH and the correlation with cell-cell fusion. However, the low pH is required for the activities of proteases in the late endosomes (such as CST-L), but not necessarily for intracellular trafficking into these compartments; and cell-cell fusion is not completely dependent on the plasma membrane fusion. For example, although the VSV-G-mediated viral entry is completely through the endocytic pathway, it still triggers effective cell-cell fusion (PMID: 30006542). In addition, it was also shown that SARS2-S-mediated cell-cell fusion requires the endosomal CTS-L (PMID: 34937699). We have discussed these issues in Discussion. It has been shown that TMPRSS2 primes SARS2-S much more effectively, resulting in more than 100-fold increase of viral infection. Thus, once spike proteins are processed on the cell surface by TMPRSS2, they could still enter these late endosomes at mildly low pH (~pH 6.7) to complete the membrane fusion and viral entry, independent of CTS-L. This would explain why TMPRSS2 reduces the low pH-dependency of SARS2 entry, which has been shown by Koch *et.al.* (PMID: 34159616), Kreutzberger *et.al.* (PMID: 36048924), and Ou *et.al.* (PMID: 33465165).

Minor Comments

1. Lines 41-42: Edit to inform that paxlovid is FDA approved and molnupiravir is approved under an Emergency Use Authorization (EUA).

This has been corrected.

2. Line 58: edit: ‘reported’ vs. ‘shown’. (The acid pH requirement has only been shown in one study (Ref. 4), and other studies support pH-independent fusion for SARS2

This has been corrected.

3. Ref. 20, which showed that U18666a and KO of NPC1 inhibit SARS2 infection, should be mentioned in the Introduction. For example it would be appropriate for Line 70 to read: “and confirmed findings (Ref. 20) indicating that entry is dependent on NPC1

The text has been revised to properly acknowledge the work done in this paper, which can be found in the beginning of Discussion.

4. Line 110: edit: “which may explain their less potent...”

It has been corrected.

5. Fig. 2B: edit: ‘Cholesterol accumulation’ (vs. ‘cholesterol trafficking’)

It has been corrected.

6. Line 119: edit: ‘by interfering with the function of NPC1’ (vs. ‘by targeting NPC1’)

It has been corrected.

7. To further assess the mechanism of tubeimosides vs. Ebola, the authors should test if tubeimosides block arrival of EBOV-GP particles to late endosomes (co-localization w/late endosome markers) and if they block at the level of fusion using viral-like particles (VLPs) tagged with beta-lactamase (e.g., as in Ref. 12).

We are interested to further elucidate the anti-Ebola activity of Tubeimosides. However, we feel these studies are not directly associated with the current work, which is heavily focusing on SARS-CoV-2. Thus, we will publish these studies in another manuscript once we collect publishable results.

8. Line 125: edit: 'EBOV-GP pseudovirus infection' (vs. EBOV infection)
It has been corrected.

9. Line 148: 'Tam' (vs. 'Tub')?
It has been corrected.

10. Fig. 3E-H legend: specify whether (as per Methods) RIPA or M2 lysis buffer was used to prepare the cell lysates. Standard RIPA buffers will denature many proteins, and so it would be preferable to prepare cell lysates for all of these experiments in a non-denaturing cell lysis buffer (e.g., the M2 lysis buffer, which the reader presumes is non-denaturing).

We apologize for the confusion. As suggested by this reviewer, the lysis buffer has been specified in Methods for these experiments. For detection of protein-protein interaction, we did not use RIPA or M2 buffer. Instead, we used this lysis buffer:

50 mM Tris, pH 7.5,
150 mM NaCl,
5% glycerol,
1 % N-Dodecyl- β -D-maltoside (DDM),
EDTA-free Protease Inhibitor.

11. RE: Figs. 3H and 5D: As above, what detergent was used to lyse the cells to purify the NPC1 protein? If RIPA, NPC1 may have been denatured during the purification. Also, whatever detergent was used for the purification, being a 13-pass transmembrane protein, NPC1 is likely not in a native state on the beads used for these experiments, which were washed 5-times with PBS.

As shown in the above, the detergent we used is DDM, which is widely used to purify multi-pass membrane proteins, such as the HIV-1 restriction factor SERINC5 in our *Cell Reports* paper (PMID: 34380030).

12. RE: Fig. 4: (a) The data in Fig. 4D should be quantitated. Ideally, the authors would perform an inherently more quantitative cell-cell fusion assay (e.g., one employing split luciferase); (b) Why are the background levels in the CMFDA channel variable? (c) Most importantly: the authors should comment on why KO of NPC1, an endosomal protein, should inhibit cell-cell fusion, which is a cell surface phenomenon.

(a) As suggested by this reviewer, **Fig.4D** has been quantified.

(b) We apologize for the poor quality of the CMFDA channel. These experiments were conducted during COVID-19 shutdown when we had very limited access to all equipment.

To further address these concerns, we repeated the cell-cell fusion experiment and new results are shown in **Fig.5I**. We now show clear images for the CMFDA channel and confirm that TMPRSS2 promotes the cell-cell fusion and chloroquine inhibits the cell-cell fusion, as reported by the others (PMID: 33051876; PMID: 34937699). These results validate our cell-cell assay. Importantly, we also show that chloroquine inhibits the cell-cell fusion when TMPRSS2 is expressed.

(c) The cell-cell fusion assay is not a direct measurement of virus-cell fusion on the plasma membrane, because the cell-cell fusion can also be triggered by the endocytic pathway. For example, although the

VSV-G-mediated viral entry is completely through the endocytic pathway, it still triggers effective cell-cell fusion (PMID: 30006542). In addition, it was also shown that SARS2-S-mediated cell-cell fusion requires the endosomal CST-L (PMID: 34937699). Thus, it is consistent that NPC1 promotes cell-cell fusion if it encounters SARS2-S.

13. RE: the effects of tubeimosides on Ebola GPcl interaction with NPC1. This would be more incisively demonstrated using purified NPC1 C-loop and purified Ebola GPcl trimeric ectodomain, e.g., in an ELISA as in Ref. 8, with analogous experiments performed for the effects of tubeimosides on an interaction between SARS2S-RBD and NPC1 C-loop.

As suggested by this reviewer, we have used purified SARS2-S RBD and NPC1-C to detect their interaction by ELISA. These new results are shown in **Fig.5E** and **Fig.5F**. We also show that this interaction is blocked by Tubeimoside III in **Fig.5G**. We feel that to further elucidate the anti-Ebola activity of tubeimosides is beyond the scope of this manuscript, but we are very pleased to explore the anti-Ebola activity of tubeimosides in future.

14. There is also some concern re: interactions reported based on co-IP experiments from 293T cells exogenously expressing NPC1 and viral glycoproteins due to likely over-expression of these proteins and questions about whether they maintain native states (e.g, for the SARS1 and MERS S experiments in Fig. 6C)

In Fig.6C, SARS1-S and MERS-S were expressed as in Fig.6A and Fig.6B to produce HIV-1 pseudovirions for infection. These pseudoviruses exhibited very high infectivity in these experiments, which were inhibited by Tubeimosides and *NPC1-KO*. Thus, it is unlikely that they were not under native states.

15. Fig. 6H: It is curious that tubeimosides and NPC1 KO increase influenza infection (Figs. 6H, I). The authors should comment on why they think this is so.

We thank this reviewer for this great question. We speculate that these influenza viruses may enter cells via NPC1-negative late endosomes, which are suppressed by NPC1.

16. L302: “low” (vs. “high”)?

It has been corrected.

17. L313-314: Clarify: Ref. 35 showed that ACE2 is not required for fusion when an alternate means of attaching SARS2S particles to target membranes is provided.

It has been clarified as suggested by this reviewer.

18. Lines 330-331: potential therapeutic utility of tubeimosides. What is known about the pharmacology of tubeimosides? Can they be given orally? By any route known, are exposure levels (e.g., Cmax and/or AUC) high enough to have an anti-viral effect. This information is needed to support therapeutic utility. If not known, the authors should delete the word ‘great’ on Lines 28 and 330.

“great” has been deleted from this sentence in both places.

Reviewer #2 (Remarks to the Author):

The manuscript entitled “SARS-CoV-2 requires Niemann-Pick C1 (NPC1) to complete cell entry” by Khan et al., presents a logical progression of experiments that cumulatively demonstrate that the host factor NPC1 is involved in SARS-CoV-2 entry. There are strengths and weaknesses to consider.

We thank this reviewer for this great comment. We have revised this manuscript based on this reviewer’s suggestions.

Major Comments:

1) One perceived weakness is that there are previous published studies that make the conclusion that NPC1 is a SARS-CoV-2 entry factor. That being said, the current study does provide the most extensive analysis. And further, based on this extensive analysis, including the first evidence of direct interaction of SARS-CoV-2 Spike and NPC1, the current report does propose a different mechanism/role for NPC1 than previous studies (which propose a more indirect role of NPC1 in regulating cholesterol levels, ACE2 levels, and even perhaps co-localizing with late endosomes that have more protease activity) (refs 1-4 below). Thus, the current study is novel in the mechanistic conclusions drawn.

We thank for these encouraging comments. We have included these references in this revision, which can be found in the beginning of Discussion.

Specific suggestions:

a. It is important for the authors to highlight the different mechanism proposed by others and discuss how their data fits into the pre-existing literature.

We appreciate this suggestion. We have expanded the Discussion section to discuss our results in comparison with the others.

b. It would be particularly compelling for the authors to test in their hands and in their experimental models, the mechanisms proposed by others (e.g., does adding back exogenous cholesterol reverse the effects on viral entry as reported, ref.1)

As suggested by this reviewer, we tested how exogenous cholesterol affects the SARS2 infection using HIV-1 pseudoviruses. We found that it slightly increases the SARS2 infectivity in wild-type cells, but it did not rescue the viral infection in *NPC1*-KO cells. This result excludes that NPC1 acts indirectly via cholesterol to promote the viral infection, which is in line with our results that NPC1 binds RBD and directly promotes the viral entry. This new result is presented in **Fig.S14**.

2) The conclusion that TMPRSS2-dependent entry relies on late endosomes (Line 245, Fig. 7, and discussion) is not supported.

a. The data in Fig. 4F could be explained the other indirect mechanisms proposed for NPC1 involvement in SARS CoV-2 entry which is unrelated to endocytosis or late endosomes, but rather changes in cholesterol at the plasma membrane and/or reduction in ACE2 levels. Considering previous reports, it is important to consider these alternative explanations before suggesting the well documented TMPRSS2 pathway has an endocytic component.

Although we appreciate this comment, our new results in **Fig.S14** do not support that NPC1 should act indirectly via cholesterol to promote the viral entry.

b. The data in Fig. 5E shows that the effect of acidification inhibitors has less effect on SAR-CoV-2

than it does on Ebola and VSV-G, which is consistent with one SARS CoV 2 entry pathway independent of endocytosis.

As we replied to Reviewer #1 (page 2, top paragraph), we respectfully propose that SARS2 enters cells via late endosomes, but not the direct breakthrough of the plasma membrane by fusion. It has been consistently shown that TMPRSS2 increases SARS2 infection more than 100-fold (PMID: 32165541; PMID: 33465165). These results demonstrate that TMPRSS2 primes SARS2-S much more effectively than CTS-L, which makes the SARS2 entry much less dependent on CST-L than Ebola and VSV. Because these acidification inhibitors selectively target proteases in the late endosomes, they have a much stronger effect on Ebola and VSV than SARS2.

3) Throughout, key information and statistical analysis is missing from the data/figures.

a. No analysis is included indicating which effects are statistically significant (i.e., p values)

As suggested by this reviewer, we completed statistical analysis by providing *p* values in all relevant figures.

b. While often the unique data points for experiments are shown, this is not always the case, and the dots are not defined (e.g., are these technical replicates or biological replicates?). It needs to be clear how many biological replicates were included in each experiment, how many times each experiment was repeated, and what the apparent “error bars” represent. For example:

- Fig. 1C - It looks like there are two types of “error bars”, what are each of these showing?

We apologize for the confusion. All experiments were repeated two or three times and error bars represent the standard error of measurements (SEMs) calculated from these replicates. The repeat (*n*) numbers have been specified for these experiments. In Fig.1C, the HIV-Luc/EBOV-GP and EBOV-trVLP data were from three independent experiments. The HIV-GFP/EBOV-GP data were from two experiments and repeated again in Fig.1D.

- Fig. 1E - What does the line represent? Is this the average of the data points shown? Are these data points biological replicates or from 3 different experiments?

We apologize for the confusion. In Fig.1E, the line indicates that all these experiments were done with HIV-1 (Luc)-EBOV-GP pseudovirions. Results from Vero cells were from three experiments, and the others were from two experiments.

Minor Comments:

1) What are the 19 compounds that were considered to be hits in Fig. S2? It is not clear why Tub1 would have been chosen. Please mark those compounds considered hits. (“decreased” misspelled on the y-axis).

We deeply apologize for the confusion. These 19 compounds are now labelled in **Fig.S2**. Tub I was selected because it showed a very strong anti-EBOV activity from this screening. The misspelling has been corrected.

2) Fig. S4, there appears to be a dose-dependent inhibition of VSV (although perhaps p values would indicate the differences are not significant. Perhaps this is related to the alternative mechanism proposed for NPC1 involving cholesterol dysregulation. If the inhibition is significant, can the authors please acknowledge this and comment on this? (Can this inhibition be countered by addition of cholesterol?)

We thank this reviewer for raising this concern. We have repeated the VSV experiment in **Fig.S4** and confirmed that these differences are not significant. These new results are now provided in the new **Fig.S4** in this revision.

3) Some things are overstated and could/should be tempered.

a. Line 38 – the CoVID pandemic has not caused “unprecedented” damages. Most previous pandemics cause much more death when the population was in fact much smaller.

“unprecedented” has been removed from this sentence.

b. Line 40 – the emergency of SARS CoV-2 VOC is not “frequent.” It is relatively slow compared to other RNA viruses

“frequent” has been removed from this sentence.

Reviewer #3 (Remarks to the Author):

Previous CRISPR screens have implicated NPC1 as a host cell susceptibility factor for SARS-CoV-2 infection. This submission provides abundant additional evidence that NPC1 is used by SARS-CoV-2 to enter cells. The findings start by showing that pharmacologic inhibitors block SARS-CoV-2 and EBOV pseudovirus entry. Knowing that EBOV GP uses NPC1 for entry, the studies then follow those used previously by filovirologists in several studies of virus-NPC1 interactions. Several important findings were made. Cells lacking NPC1 are relatively resistant to pseudovirus entry SARS-CoV-2, spikes bind NPC1, binding is blocked by the inhibitors. The findings with SARS-CoV-2 pseudoviruses were extended to authentic SARS-CoV-2, which was also blocked by inhibitors and was also dependent on NPC1 for infection. The findings were also extended to SARS-CoV and to MERS-CoV, which also showed some dependence on NPC1 for entry. These are significant and important findings for the field.

We would like to thank this reviewer for these great comments.

Comments:

1. There are concerns about the support for statements that SARS-CoV-2 is “completely” dependent on endocytosis (line 26, lines 253-254 and elsewhere). The dependence for endosomal NPC1 would support this statement, but the results with inhibitors of endosomal acidification (Fig 5E) do not. Also, the result in Fig 4F is very interesting but cannot be used to conclude that SARS-CoV-2 entry is exclusively endosomal. The discussion on lines 295-305 might be further developed to incorporate findings on the relationships between SARS-CoV-2 spike cleavage and entry kinetics. There are several reports demonstrating that prior cleavage of SARS-CoV-2 spikes will allow for bypass of endosomal neutralizing agents and cell-surface CoV entry.

As we replied to Reviewer #1 (page 2, top paragraph) and Reviewer #2 (page 6, top paragraph), we respectfully propose that SARS2 enters cells via late endosomes, but not the direct breakthrough of the plasma membrane by fusion. As pointed out by this Reviewer, the furin-cleave makes the priming of SARS1-S and SARS2-S less dependent on CTS-L, resulting in the viral entry become less sensitive to endosomal neutralizing agents. However, the bypass of these agents should not suggest that the virus shifted from endosomes to the cell-surface for membrane fusion and entry. Instead, it only indicates that the priming by protease is dependent rather on TMPRSS2 than on CTS-L. We have discussed these issues in Discussion.

2. Extending from point 1, is it not puzzling that NPC1 facilitates cell-to-cell fusion (Fig 4D)? How is this finding reconciled with the acid pH dependence and endosomal spike-mediated fusion process that is promoted in the manuscript?

As we replied to Reviewer #1 (page 4, top paragraph), the cell-cell fusion can also be triggered by viruses such as VSV which is completely dependent on the endocytic entry. In addition, SARS2-S also mediates cell-cell fusion by this mechanism, which is confirmed by our new results in Fig.5I. Thus, our results are consistent with these previous findings.

3. Discussion of the findings in conjunction with a related recent paper (<https://doi.org/10.1016/j.cell.2023.06.005>) would add impact, as this paper highlights a separate endolysosomal transmembrane protein in facilitating SARS-CoV-2 cell entry.

As suggested by this review, we have discussed the new finding of TMEM106B on SARS2 entry in this revision in Discussion.

4. Additional consideration of the model in Fig 7 may be helpful. SARS-CoV-2 entry required ACE2, a well-known receptor that drives entry through conformational changes in spike proteins. Findings in this submission showed that SARS-2 entry was facilitated by NPC1, but only when ACE2 was also present. The discussion on lines 310-314 give the impression that a single spike trimer may bind a single ACE2 monomer, leaving two of three trimer subunits available for NPC1 binding. If this is the model, it could be made more apparent in Fig 7, which at present appears underdeveloped in relation to the findings of the paper. The model also should account for the fact that there are many SARS-CoV-2 attachment factors in addition to ACE2 that are known to bring SARS-CoV-2 into endosomes, so ACE2 is not liable to be a mere “attachment factor”, given its essential requirement for entry in all experiments of this submission.

We developed this model from our results and the reference (PMID: 36971081) that only one S protein binds an ACE2 for viral entry. As this reviewer pointed out, there are many entry co-factors for SARS-CoV-2 have been reported, but their role needs to be further validated and acknowledged in the field. Thus, we wish to present a simple model that only includes those well-accepted host proteins (ACE2, TMPRSS2, CTS-L) to illustrate our findings. We have discussed the difference of ACE2 with these attachment factors in Discussion, to address these concerns.

Additional minor comments:

1. Consider leading introduction with refs 38, 39, as these findings provide a premise for focusing on NPC1.

We wish to share with this Reviewer that our work on NPC1 was not initiated because of these two papers. We did not publish our work earlier because we wanted to firmly confirm the NPC1 function by CRISPR/Cas9 knockout and elucidate the underlying mechanism, which took us a lot of time. Nonetheless, we agree that it is important to discuss their work in the context of our findings, which is described in the beginning of Discussion.

2. Consider sticking with the historical nomenclature (SARS-CoV-2, not “SARS2”), although this may be decided by the journal and its standards for consistent nomenclature.

We appreciate this suggestion. We have avoided using “SARS2” in Title, Abstract, Introduction, and Discussion, and only used it in our data presentation.

3. Line 48, mutations beyond S gene also produce VOC.

This sentence has been revised accordingly.

4. Line 53, ACE2 does drive conformation changes; this correct statement appears at odds with lines 315-316, which appear to demote ACE2 to a mere “attachment factor”. Some reconsideration of the roles for ACE2 in relation to NPC1 would be helpful in the discussion.

We have revised the text in Discussion to discuss more carefully on the role of ACE2 in SARS2 infection.

Reviewers' Comments:

Reviewer #1 (Remarks to the Author):

The authors have revised their manuscript, which reports an interaction between SARS-CoV-2 S and NPC1 and its blockade by tubeimosides. While they have added new data and clarified some points, the text, in particular of the Discussion, needs further amendments by attending to the following comments.

MAJOR COMMENTS

RE: Original Major Comments (Numbering as per original review # 1.)

1. RE: Original Main Comment 1: The authors contend a direct and consequential interaction between SARS-CoV-2 S and NPC1. In response to requests for further substantiation, the authors added ELISA data (new Figs. 6FG). While helpful, the authors should acknowledge (e.g., append to Line 380) that future work is required to visualize the SARS-CoV-2 S--NPC1 interaction and to show that it is functionally relevant to fusion.

2. RE: Original Main Comment 2: The authors propose that irrespective of cell type (i.e., whether the cell expresses TMPRSS2 at the surface), SARS-CoV-2 requires endocytosis to LATE endosomes and binding between SARS-CoV-2 S and NPC1 (a LATE endosome protein) to trigger fusion and entry (i.e., that this is the sole mechanism of SARS-CoV-2 entry into all cells. In response to concerns that this is at odds with information in the literature, the authors offered several arguments. But there are caveats to some, and the Discussion should be tempered to reflect that entry through NPC1+ late endosomes is likely not the only pathway for SARS-CoV-2 entry into all cell types. For example, the late endosomal, NPC1-dependent pathway may not be needed in physiologically relevant cells, such as those of the nasal cavity and upper respiratory tract.

(i) The authors cite Ref. 4 as evidence and state that SARS-CoV-2 needs to pass through late endosomes for entry (on L66-67, L355-356 and elsewhere). But Ref. 4 indicates that there are 4 potential pathways for SARS-CoV-2 entry: (a) directly from the cell surface in, for example, cells of the nasal cavity where a pH of ~6.7 was recorded; (b) in early (NPC1-negative) endosomes in TMPRSS2+ cells (pH ~6.0 to 6.8); (c) in late (NPC1+) endosomes (pH \leq 5.5 to 6) in TMPRSS2+ cells, and (d) in late endosomes in TMPRSS2-negative cells (i.e., for cathepsin cleavage). Ref. 4 also reports that only mildly low pH (~6.8) is needed for SARS-CoV-2 S fusion triggering, which would prevail in two of the NPC1-negative entry locales: (a) at the cell surface (in the nasal cavity) and (b) in early (NPC1-negative) endosomes in TMPRSS2+ cells (i.e., in the upper respiratory tract). If fusion can occur in NPC1-negative compartments (as Ref. 4 demonstrated with live cell microscopy at the plasma membrane in response to pH 6.8 and in early endosomes that are NPC1-negative), then it is hard to reason that NPC1 is an obligate SARS-CoV-2 entry factor for all entry pathways. Perhaps NPC1 is indeed needed, perhaps through a direct interaction with SARS-CoV-2 S, in two (of the above four) situations: when a SARS-CoV-2 particle hasn't yet fused at the plasma membrane or in an early endosome of a TMPRSS2+ cell, or if, due to lack of TMPRSS2, it must traffic to late endosomes so that cat L can perform the obligate S2' site cleavage.

(ii) RE: the query re: Fig. 4D as to why SARS-CoV-2 S-mediated cell-cell fusion should require NPC1 (a late endosomal protein), the authors respond that cell-cell fusion requires cathepsin L (cat L, a late endosomal protein), citing PMID 34937699. But PMID 34937699 shows that cell-to-cell transmission, not cell-to-cell fusion, requires cat L. (See PMID: 37591996 for a recent paper on cell-to-cell transmission (of an alphavirus), without apparent cell-to-cell fusion).

(iii) Fig. 6H (effects of NH₄Cl and chloroquine on infection). The data support a low pH requirement for entry but do not indicate whether entry is through early endosomes (NPC1-negative) or late endosomes (NPC1-positive). The observation that VSV-G and Ebola GP entry are more sensitive to the lysosomotropic agents than SARS-CoV-2 S is consistent with SARS-CoV-2 fusing at a higher pH than Ebola GP, and at even a higher pH than VSV G, which induces fusion at pH 6. Data in Fig. 6H are therefore more in accord with entry through early, rather than late,

endosomes.

NEW MAJOR COMMENTS

3. L320: Add to the end of sentence: "in monkey kidney cells". Ref. 34 was performed in BSC-1 grivet kidney cells, which like Vero cells likely do not express TMPRSS2.
4. Lines 348-350 should be edited or deleted. It is well-accepted that VSV G only induces cell-cell fusion if the pH is lowered to 6 or below. The Methods section of Ref. 44 (PMID: 30006542) states that the cells were treated for 1 min at pH 6 or pH 5 (to trigger fusion). Fig. 1 and its legend (of Ref. 44) denote the low pH step. All other Figures (or their legends or description of Results) documenting VSV G-mediated cell-cell fusion indicate that low pH was employed as the fusion trigger. Hence this is not an apt comparison of the situation with SARS-CoV-2, which the authors contend always enters all cells through late endosomes, yet can cause cell-cell fusion at neutral pH.
5. Lines 350-351 should be edited or deleted. As per Major Original comment 2 (point (ii)): Ref. 32 shows that cell-to-cell transmission, not cell-to-cell fusion, depends on cat L.
6. Lines 352-354 should be edited. Ref. 4 did show direct fusion at the plasma membrane, when the medium pH was lowered to 6.8, which the authors (of Ref. 4) contend mimics the situation in the nasal cavity.
7. Lines 354-356 should be edited. Ref. 4 found that in TMPRSS2+ cells about 50% of fusion events occurred in early endosomes, and ~50% occurred in late endosomes (see Supplemental Fig. 15).

(NEW) MINOR COMMENTS

1. Have the Fig. #s for Figs. 5 and 6 been swapped in the text? If so, correct.
2. L325: Edit. The reduction was about 3X in Ref. 20 for the NPC1 KO cells.

Reviewer #2 (Remarks to the Author):

The manuscript now entitled "SARS-CoV-2 entry is dependent on Niemann-Pick C1 (NPC1) and blocked by Tubeimosides" by Khan et al., has been revised in response to Reviewer comments. While some issues have been addressed, a major point raised by all 3 Reviewers was not addressed (Major point 1 below).

Major Comments:

- 1) All 3 Reviewers question the main conclusion of the manuscript that all SARS-CoV-2 entry is dependent on endocytosis (and NPC1). Looking at the data presented, the Reviewers unanimously saw the results as consistent with the literature that there are two distinct entry pathways for SARS-CoV-2 because inhibition of NPC1 nor neutralization of pH changes completely inhibit SARS-CoV-2 entry as seen for EBOV. In response, the authors have repeated their alternative interpretation, but this does not address the issue in a tangible way. The authors can certainly propose an alternative interpretation (and explain their speculation that the lack of complete inhibition is related to TMPRSS2 digestion making Spike less dependent on CTS-L), but with the data presented they would have to acknowledge that the entry that is not completely inhibited by their treatments and that the remaining entry observed could indeed be plasma membrane mediated. But for now, this major critique of the study conclusions remains unaddressed.
- 2) As requested, the authors have added numerous references that previously have reported NPC1 as a SARS-CoV-2 entry factor (e.g., 20, 34, 35, 36, 37, 38, 39, 40), however, the title of the manuscript still suggests this is a new discovery. The authors certainly have extended the finding about NPC1 being involved in SARS-CoV-2 entry, but the story the authors are telling in the manuscript does not align well with the title. New findings that one might consider highlighting in

the title might be the evidence that Spike binds NPC1, the demonstration of NPC1 as an effective target for SARS-CoV-2, and specifically Tubeimosides (and other NPC1 inhibitors) as pan-CoV and filovirus inhibitors (particularly relevant to the continued evolution of SARS-CoV-2 and the remaining pandemic threat presented by CoV and filoviruses).

3) I apologize for not seeing this before, but there seems to be an inconsistency with the SARS2-S cleavage data in Figure 3.

a. In the right panel of figure 3H Spike in the absence of the TL protease shows bands around 250kd and slightly below 100kd. The cleaved samples (in the presence of the TL protease) in the same panel show 3 Spike bands, one above 150kd but below 250kd, one just below 100kd, and one at 70kd.

However, in panel J the Spike WB patterns are different. The large band in the Fig.3J top right panel which includes the TL protease shows the highest band at above 250kd (consistent with the uncleaved samples in 3H). The top band in the Fig. 3J bottom right panel which does not include the protease shows the highest band above 150kd but below 250kd (consistent with the cleaved samples in 3H). No S band should be larger in the cleaved samples, so it seems that there is something wrong with the gels in Fig. 3J.

b. Can the author please clarify what bands are expected under both conditions and ensure the bands they have observed are consistent with those expectations and explain why there are differences between their different blots.

4) Reviewer 1 suggested that information about pharmacology of tubeimosides be included, and a bit has been published about tubeimosides I, II, and III in this regard. It would be helpful to include this information to support the conclusion that these are promising antivirals. They have been studied in vivo, so the data available is supportive of these claims.

Minor/Specific Comments:

1) There are English issues throughout the manuscript

2) The abstract is a bit repetitive (specifically, lines 23-26)

3) The authors have not shown that NPC1 is "indispensable" for all SARS-CoV-2 entry as stated in the abstract. NPC1 KO does not prevent SARS-CoV-2 entry as completely as it did for EBOV.

4) I may have missed it, but do not find the CTS-B and CTS-L abbreviations defined in the manuscript

5) Was anything done to confirm the conformation of the NPC1L1 deletion mutants 1-337 and 1-620?

6) Figure 4 says D786N instead of D787N.

7) Fig. 4a –define the different colored peaks in the flow graph.

8) Authors need to discuss why in their time of addition studies inhibition was observed 2 hours post-infection. Were the samples not synchronized via spinoculation or some other method?

9) In Figure 3F it looks like some full length NPC1 is present in the mutant lanes. Please address this.

10) Fig. 4d there is so much CMFDA background everything looks yellow in the merge. It is indicated these syncytia were counted by an automated program. How does the machine count syncytia, by size or color? And the numbers presented in the graph, is this from representative fields or replicate wells?

Reviewer #3 (Remarks to the Author):

Second round of review

The submission is improved, yet some questions remain.

1. Lines 264-281; "TMPRSS2 requires acidic environment"; this reviewer still questions whether the findings support the conclusions. SARS2 spike entry is significantly more resistant to chloroquine and ammonium chloride than EBOVGP and VSVG entry. TMPRSS2 reduces sensitivity to these bases. That chloroquine reduces TMPRSS2-facilitated cell-cell fusion does not "suggest that TMPRSS2 should not shift SARS2 entry from the endocytic route". Chloroquine is pleiotropic; it could reduce spike cleavage or cell surface presentation, thus reducing syncytia.
2. Discussion, lines 347-350; that VSVG can cause minor syncytia is not an argument for endosome involvement in plasma membrane fusions. It is hard to buy the arguments put forward on lines 347-350. That CTS-L might increase syncytia can be explained by cell surface CTS-L, which is well known.
3. Discussion lines 336-362; the TMPRSS2 and CSTL are activating proteases cleaving at S2prime, not priming proteases as stated in discussion and elsewhere. While the report is thorough, it does not align NPC1 binding to spike with well described ACE2-induced conformational changes in spikes, spike cleavages at S1-S2 (the priming cleavages) and spike cleavages at S2prime (the activating cleavages). This could be stated in discussion.

Reviewers' comments:

Reviewer #1 (Remarks to the Author):

The authors have revised their manuscript, which reports an interaction between SARS-CoV-2 S and NPC1 and its blockade by tubeimosides. While they have added new data and clarified some points, the text, in particular of the Discussion, needs further amendments by attending to the following comments.

MAJOR COMMENTS

RE: Original Major Comments (Numbering as per original review # 1.)

1. RE: Original Main Comment 1: The authors contend a direct and consequential interaction between SARS-CoV-2 S and NPC1. In response to requests for further substantiation, the authors added ELISA data (new Figs. 6FG). While helpful, the authors should acknowledge (e.g., append to Line 380) that future work is required to visualize the SARS-CoV-2 S--NPC1 interaction and to show that it is functionally relevant to fusion.

Thanks for this comment. This acknowledgement has been included in Discussion as suggested.

2. RE: Original Main Comment 2: The authors propose that irrespective of cell type (i.e., whether the cell expresses TMPRSS2 at the surface), SARS-CoV-2 requires endocytosis to LATE endosomes and binding between SARS-CoV-2 S and NPC1 (a LATE endosome protein) to trigger fusion and entry (i.e., that this is the sole mechanism of SARS-CoV-2 entry into all cells. In response to concerns that this is at odds with information in the literature, the authors offered several arguments. But there are caveats to some, and the Discussion should be tempered to reflect that entry through NPC1+ late endosomes is likely not the only pathway for SARS-CoV-2 entry into all cell types. For example, the late endosomal, NPC1-dependent pathway may not be needed in physiologically relevant cells, such as those of the nasal cavity and upper respiratory tract.

The Discussion has been revised to include additional entry routes as suggested (also see the new model in Fig.7).

(i) The authors cite Ref. 4 as evidence and state that SARS-CoV-2 needs to pass through late endosomes for entry (on L66-67, L355-356 and elsewhere). But Ref. 4 indicates that there are 4 potential pathways for SARS-CoV-2 entry: (a) directly from the cell surface in, for example, cells of the nasal cavity where a pH of ~6.7 was recorded; (b) in early (NPC1-negative) endosomes in TMPRSS2+ cells (pH ~6.0 to 6.8); (c) in late (NPC1+) endosomes (pH ≤ 5.5 to 6) in TMPRSS2+ cells, and (d) in late endosomes in TMPRSS2-negative cells (i.e., for cathepsin cleavage). Ref. 4 also reports that only mildly low pH (~6.8) is needed for SARS-CoV-2 S fusion triggering, which would prevail in two of the NPC1-negative entry locales: (a) at the cell surface (in the nasal cavity) and (b) in early (NPC1-negative) endosomes in TMPRSS2+ cells (i.e., in the upper respiratory tract). If fusion can occur in NPC1-negative compartments (as Ref. 4 demonstrated with live cell microscopy at the plasma membrane in response to pH 6.8 and in early endosomes that are NPC1-negative), then it is hard to reason that NPC1 is an obligate SARS-

CoV-2 entry factor for all entry pathways. Perhaps NPC1 is indeed needed, perhaps through a direct interaction with SARS-CoV-2 S, in two (of the above four) situations: when a SARS-CoV-2 particle hasn't yet fused at the plasma membrane or in an early endosome of a TMPRSS2+ cell, or if, due to lack of TMPRSS2, it must traffic to late endosomes so that cat L can perform the obligate S2' site cleavage.

We have discussed these entry pathways from Ref #4 and balanced them in Discussion. We also revised our model (Fig.7).

(ii) RE: the query re: Fig. 4D as to why SARS-CoV-2 S-mediated cell-cell fusion should require NPC1 (a late endosomal protein), the authors respond that cell-cell fusion requires cathepsin L (cat L, a late endosomal protein), citing PMID 34937699. But PMID 34937699 shows that cell-to-cell transmission, not cell-to-cell fusion, requires cat L. (See PMID: 37591996 for a recent paper on cell-to-cell transmission (of an alphavirus), without apparent cell-to-cell fusion).

This part in Discussion has been deleted as suggested (see below).

(iii) Fig. 6H (effects of NH₄Cl and chloroquine on infection). The data support a low pH requirement for entry but do not indicate whether entry is through early endosomes (NPC1-negative) or late endosomes (NPC1-positive). The observation that VSV-G and Ebola GP entry are more sensitive to the lysosomotropic agents than SARS-CoV-2 S is consistent with SARS-CoV-2 fusing at a higher pH than Ebola GP, and at even a higher pH than VSV G, which induces fusion at pH 6. Data in Fig. 6H are therefore more in accord with entry through early, rather than late, endosomes.

We thank for these comments. The Discussion has been revised to include these discussions.

NEW MAJOR COMMENTS

3. L320: Add to the end of sentence: "in monkey kidney cells". Ref. 34 was performed in BSC-1 grivet kidney cells, which like Vero cells likely do not express TMPRSS2.

It has been added to this sentence.

4. Lines 348-350 should be edited or deleted. It is well-accepted that VSV G only induces cell-cell fusion if the pH is lowered to 6 or below. The Methods section of Ref. 44 (PMID: 30006542) states that the cells were treated for 1 min at pH 6 or pH 5 (to trigger fusion). Fig. 1 and its legend (of Ref. 44) denote the low pH step. All other Figures (or their legends or description of Results) documenting VSV G-mediated cell-cell fusion indicate that low pH was employed as the fusion trigger. Hence this is not an apt comparison of the situation with SARS-CoV-2, which the authors contend always enters all cells through late endosomes, yet can cause cell-cell fusion at neutral pH.

These two lines have been deleted as suggested.

5. Lines 350-351 should be edited or deleted. As per Major Original comment 2 (point (ii)): Ref. 32 shows that cell-to-cell transmission, not cell-to-cell fusion, depends on cat L.

These two lines have been deleted as suggested.

6. Lines 352-354 should be edited. Ref. 4 did show direct fusion at the plasma membrane, when the medium pH was lowered to 6.8, which the authors (of Ref. 4) contend mimics the situation in the nasal cavity.

These two lines have been deleted as suggested.

7. Lines 354-356 should be edited. Ref. 4 found that in TMPRSS2+ cells about 50% of fusion events occurred in early endosomes, and ~50% occurred in late endosomes (see Supplemental Fig. 15).

These two lines have been deleted as suggested.

(NEW) MINOR COMMENTS

1. Have the Fig. #s for Figs. 5 and 6 been swapped in the text? If so, correct.

It has been corrected. These two figures were swapped during uploading.

2. L325: Edit. The reduction was about 3X in Ref. 20 for the NPC1 KO cells.

It has been corrected to be "3X".

Reviewer #2 (Remarks to the Author):

The manuscript now entitled “SARS-CoV-2 entry is dependent on Niemann-Pick C1 (NPC1) and blocked by Tubeimosides” by Khan et al., has been revised in response to Reviewer comments. While some issues have been addressed, a major point raised by all 3 Reviewers was not addressed (Major point 1 below).

This major point is addressed in this revision.

Major Comments:

1) All 3 Reviewers question the main conclusion of the manuscript that all SARS-CoV-2 entry is dependent on endocytosis (and NPC1). Looking at the data presented, the Reviewers unanimously saw the results as consistent with the literature that there are two distinct entry pathways for SARS-CoV-2 because inhibition of NPC1 nor neutralization of pH changes completely inhibit SARS-CoV-2 entry as seen for EBOV. In response, the authors have repeated their alternative interpretation, but this does not address the issue in a tangible way. The authors can certainly propose an alternative interpretation (and explain their speculation that the lack of complete inhibition is related to TMPRSS2 digestion making Spike less dependent on CTS-L), but with the data presented they would have to acknowledge that the entry that is not completely inhibited by their treatments and that the remaining entry observed could indeed be plasma membrane mediated. But for now, this major critique of the study conclusions remains unaddressed.

We thank for these comments. We have revised the text in Discussion to include NPC1-independent entry routes for SARS-CoV-2 (also see the new model in Fig.7).

2) As requested, the authors have added numerous references that previously have reported NPC1 as a SARS-CoV-2 entry factor (e.g., 20, 34, 35, 36, 37, 38, 39, 40), however, the title of the manuscript still suggests this is a new discovery. The authors certainly have extended the finding about NPC1 being involved in SARS-CoV-2 entry, but the story the authors are telling in the manuscript does not align well with the title. New findings that one might consider highlighting in the title might be the evidence that Spike binds NPC1, the demonstration of NPC1 as an effective target for SARS-CoV-2, and specifically Tubeimosides (and other NPC1 inhibitors) as pan-CoV and filovirus inhibitors (particularly relevant to the continued evolution of SARS-CoV-2 and the remaining pandemic threat presented by CoV and filoviruses).

The title has been changed as suggested. We appreciate this suggestion.

3) I apologize for not seeing this before, but there seems to be an inconsistency with the SARS2-S cleavage data in Figure 3.

a. In the right panel of figure 3H Spike in the absence of the TL protease shows bands around 250kd and slightly below 100kd. The cleaved samples (in the presence of the TL protease) in the same panel show 3 Spike bands, one above 150kd but below 250kd, one just below 100kd, and one at 70kd. However, in panel J the Spike WB patterns are different. The large band in the Fig.3J top right panel which includes the TL protease shows the highest band at above 250kd (consistent with the uncleaved samples in 3H). The top band in the Fig. 3J bottom right panel which does not include the protease shows the highest band above 150kd but below 250kd

(consistent with the cleaved samples in 3H). No S band should be larger in the cleaved samples, so it seems that there is something wrong with the gels in Fig. 3J.

We apologize that our labeling for the markers was inaccurate, which has been corrected. It is correct that no S band should be larger than 250 kDa in the cleaved samples.

b. Can the author please clarify what bands are expected under both conditions and ensure the bands they have observed are consistent with those expectations and explain why there are differences between their different blots.

In our hands, the undigested S_0 was more than 250 kDa, and digested S_0 was a little less than 250 kDa. We could also detect a digested 70 kDa band from S_2 .

4) Reviewer 1 suggested that information about pharmacology of tubeimosides be included, and a bit has been published about tubeimosides I, II, and III in this regard. It would be helpful to include this information to support the conclusion that these are promising antivirals. They have been studied in vivo, so the data available is supportive of these claims.

The pharmacokinetics of Tubeimoside I has been investigated in animal models, which showed a very slow clearance, with maximum concentration of $\sim 1,000$ nM at ~ 2.85 h in rat plasma after intravenous administration. This information has been included in Discussion.

Minor/Specific Comments:

1) There are English issues throughout the manuscript.

We apologize for these issues. The text has been carefully proofread to correct any English issues.

2) The abstract is a bit repetitive (specifically, lines 23-26)

The abstract has been revised to make it more concise.

3) The authors have not shown that NPC1 is “indispensable” for all SARS-CoV-2 entry as stated in the abstract. NPC1 KO does not prevent SARS-CoV-2 entry as completely as it did for EBOV.

The text has been revised to describe our results more precisely. “indispensable” has been deleted from the Abstract.

4) I may have missed it, but do not find the CTS-B and CTS-L abbreviations defined in the manuscript

CTS (cathepsin) was defined in the introduction.

5) Was anything done to confirm the conformation of the NPC1L1 deletion mutants 1-337 and 1-620?

We detected their expression by Western blotting, which showed expected sizes. In addition, the 1-620 mutant bound EBOV-GP, whereas the 1-337 mutant did not, when detected by immunoprecipitation, which is consistent with that EBOV-GP binds NPC1-C loop. These results indicate that these two proteins exhibit functional conformation.

6) Figure 4 says D786N instead of D787N.

We apologize for this typo, which has been corrected to D786N in the result.

7) Fig. 4a –define the different colored peaks in the flow graph.

The peaks are defined. The red colored graph indicates A549-WT cells that express little ACE2 (-) and the blue colored graph indicates A549-WT-A cells that express high levels of ACE2 (+).

8) Authors need to discuss why in their time of addition studies inhibition was observed 2 hours post-infection. Were the samples not synchronized via spinoculation or some other method?

We used the same protocol that was used to demonstrate that MBX2254 and MBX2270 inhibit Ebola virus entry (PMID: 26206510). This protocol does not involve synchronization or spinoculation, which may explain the inhibition 2 hours post-infection. We apologize that we did not include this reference, which is included now in the result section.

9) In Figure 3F it looks like some full length NPC1 is present in the mutant lanes. Please address this.

We detected an unspecific band from mutants 1-620 and 1-377 in this experiment. Although it showed a similar size as the full-length NPC1, it was not from the full-length protein. Otherwise, we should detect an interaction of the 1-377 mutant with GP or S proteins b IP.

10) Fig. 4d there is so much CMFDA background everything looks yellow in the merge. It is indicated these syncytia were counted by an automated program. How does the machine count syncytia, by size or color? And the numbers presented in the graph, is this from representative fields or replicate wells?

These syncytia were counted manually in live cells after visualization by IncuCyte. Syncytia were determined by cell morphology in each well from three independent experiments.

Reviewer #3 (Remarks to the Author):

Second round of review

The submission is improved, yet some questions remain.

1. Lines 264-281; “TMPRSS2 requires acidic environment”; this reviewer still questions whether the findings support the conclusions. SARS2 spike entry is significantly more resistant to chloroquine and ammonium chloride than EBOVGP and VSVG entry. TMPRSS2 reduces sensitivity to these bases. That chloroquine reduces TMPRSS2-facilitated cell-cell fusion does not “suggest that TMPRSS2 should not shift SARS2 entry from the endocytic route”. Chloroquine is pleiotropic; it could reduce spike cleavage or cell surface presentation, thus reducing syncytia. Thanks for these comments. This statement has been deleted and the text has been revised.

2. Discussion, lines 347-350; that VSVG can cause minor syncytia is not an argument for endosome involvement in plasma membrane fusions. It is hard to buy the arguments put forward on lines 347-350. That CTS-L might increase syncytia can be explained by cell surface CTS-L, which is well known.

This part of Discussion has been deleted.

3. Discussion lines 336-362; the TMPRSS2 and CSTL are activating proteases cleaving at S2prime, not priming proteases as stated in discussion and elsewhere. While the report is thorough, it does not align NPC1 binding to spike with well described ACE2-induced conformational changes in spikes, spike cleavages at S1-S2 (the priming cleavages) and spike cleavages at S2prime (the activating cleavages). This could be stated in discussion.

This part of Discussion has been deleted. We have also deleted “priming” to avoid any misunderstanding. We have included a statement that future work is required for the role of NPC1 in SARS-CoV-2 fusion (lines 353-355).

REVIEWER COMMENTS

Reviewer #1 (Remarks to the Author):

The authors have submitted a second revision of their manuscript, which is now entitled: "Tubeimosides are pan-coronavirus and filovirus inhibitors that block their fusion protein binding to Niemann-Pick C1". They have responded by amending the text and modifying the model Figure (Fig. 7). As outlined below, there are still some interpretive concerns that require attention.

Specific Comments

1. Given lingering caveats about the binding studies, as expressed in prior reviews and, e.g. the IP of full length Ebola GP with anti-NPC1 (Figs. 3E, F), and SARS2 S protein found in precipitates of samples that did not include NPC1 (Fig. 3H, right) and to reflect that tubeimosides may have alternate/additional modes of action, the authors should make a few additional minor amendments to tone down some of the claims. (a) Change the title of the manuscript to read: "Tubeimosides are pan-coronavirus and filovirus inhibitors that can block their fusion protein binding to Niemann-Pick C1"; (b) Abstract L24: "provide evidence" (vs. "demonstrate"); (c) L151: add "in certain cells" to the end of the sentence.

2. To this reader it is still a major conundrum why lysosomotropic agents should inhibit cell-cell fusion (Fig. 5I and text Lines 267-272). The authors now cite reference 33 as previously showing this. But this is a mis-reading of Ref. 33, which states: "We also applied these inhibitors* to cell-cell fusion assays but found no effect on either SARS-CoV-2 or SARS-CoV, as would be expected (SI Appendix, Fig. S3)" ["*These inhibitors" refers to cathepsin L inhibitor III, cathepsin B inhibitor CA-074, E-64d (general cathepsin inhibitor), BafA1 (ATPase pump inhibitor), and leupeptin (general protease inhibitor), BafA1 being the lysosomotropic agent employed.] Furthermore, why is the mCherry signal weaker in some of the panels, notably in the bottom 3 rows, which show samples treated with chloroquine? Perhaps low mCherry signal obscured the ability to detect syncytia; the Method was not explained in detail. Was it simply based on yellow intensity? Was there a size threshold imposed? Irrespective of the assay details, as the effect of chloroquine on cell-cell fusion remains a conundrum and since there are better quantitative cell-cell fusion assays available (e.g., split-GFP or split-luciferase complementation), this reader feels strongly that Fig. 5I and text thereof (in Results and Discussion) should be omitted.

3. Based on the same concerns about the cell-cell fusion assay, this reader is also of the opinion that Fig. 4D should be deleted and that Lines 25-26 of the Abstract should be revised, for example to read: "NPC1 strongly promotes productive SARS-CoV-2 entry, which we propose is due to its influence on fusion in late endosomes."

4. The authors should make a few more modifications to the Discussion: (a) Text between L323-329. Eliminate the terms "dominant" and "minor"; NPC1-independent entry through the plasma membrane in nasal cavity, where the extracellular pH was recorded to be ~6.7, is likely an/the most important route of infection clinically. (b) L347-348: specify that "in some cells" ACE2 is more likely an attachment factor.

5. Line 154: Add to end of sentence: ", albeit higher than the IC50s recorded for inhibition of Ebola pseudovirion infections".

6. Line 217: Delete "strongly"; the effect of NPC1 KO on SARS2 infection of Caco2 cells is less than 1 log10.

7. Line 257: Replace "it" with "mildly low pH".

8. Fig. 5I: Correct spelling of mCherry.

9. Line 292: The statement should be qualified as it looks like Omicron D18 S IP'd with the 1-377

fragment of NPC1 (in Supp. Fig. 12, IP lane 11).

10. L312: Specify the cell type for Ref 22 NPC1 KO, as cell type is critical to relative usage of plasma membrane vs. endosomal entry for SARS-CoVs. Ref. 22 used NPC1 KO 293T-ACE2-TMPRSS2 cells.

11. L351: Do you mean to say: "to initiate endocytic entry". Alternatively, say "to promote Ebola particle internalization".

Reviewer #2 (Remarks to the Author):

The authors have addressed the concerns of the reviewers. The small remaining comments are as follows:

- 1) lines 178, 219 and 317- because infection is not completely blocked NPC1 KO, it would be more appropriate to say NPC1 is involved (or a critical factor) rather than required which implies absolute.
- 2) there are still English issues, but hopefully these will be addressed during editorial review

Reviewer #1 (Remarks to the Author):

The authors have submitted a second revision of their manuscript, which is now entitled: “Tubeimosides are pan-coronavirus and filovirus inhibitors that block their fusion protein binding to Niemann-Pick C1”. They have responded by amending the text and modifying the model Figure (Fig. 7). As outlined below, there are still some interpretive concerns that require attention.

We thank you for these positive comments and we have addressed these concerns.

Specific Comments

1. Given lingering caveats about the binding studies, as expressed in prior reviews and, e.g. the IP of full length Ebola GP with anti-NPC1 (Figs. 3E, F), and SARS2 S protein found in precipitates of samples that did not include NPC1 (Fig. 3H, right) and to reflect that tubeimosides may have alternate/additional modes of action, the authors should make a few additional minor amendments to tone down some of the claims. (a) Change the title of the manuscript to read: “Tubeimosides are pan-coronavirus and filovirus inhibitors that can block their fusion protein binding to Niemann-Pick C1”; (b) Abstract L24: “provide evidence” (vs. “demonstrate”); (c) L151: add “in certain cells” to the end of the sentence. These three changes have been made in lines 1 (Title), 23-24 (Abstract), and 147 (Results).

2. To this reader it is still a major conundrum why lysosomotropic agents should inhibit cell-cell fusion (Fig. 5I and text Lines 267-272). The authors now cite reference 33 as previously showing this. But this is a mis-reading of Ref. 33, which states: “We also applied these inhibitors* to cell–cell fusion assays but found no effect on either SARS-CoV-2 or SARS-CoV, as would be expected (SI Appendix, Fig.S3)” [*“These inhibitors” refers to cathepsin L inhibitor III, cathepsin B inhibitor CA-074, E-64d (general cathepsin inhibitor), BafA1 (ATPase pump inhibitor), and leupeptin (general protease inhibitor), BafA1 being the lysosomotropic agent employed.] Furthermore, why is the mCherry signal weaker in some of the panels, notably in the bottom 3 rows, which show samples treated with chloroquine? Perhaps low mCherry signal obscured the ability to detect syncytia; the Method was not explained in detail. Was it simply based on yellow intensity? Was there a size threshold imposed? Irrespective of the assay details, as the effect of chloroquine on cell-cell fusion remains a conundrum and since there are better quantitative cell-cell fusion assays available (e.g., split-GFP or split-luciferase complementation), this reader feels strongly that Fig. 5I and text thereof (in Results and Discussion) should be omitted.

We appreciate these comments.

In Ref #33, it was shown that SARS2-S protein-mediated cell-cell fusion contributes to SARS2 cell-cell transmission, which is consistent with that cell-cell fusion plays an important role in cell-cell transmission. However, it was also shown in this paper that these lysosomotropic inhibitors only block cell-cell transmission, but have no effects on cell-cell fusion, which is surprising.

We wish to point out this reviewer that in Fig.5I, the mCherry signal from the bottom 3 rows (all chloroquine treated) is comparable to that from the top row (untreated). The mCherry signal from the 2nd and 3rd rows is stronger than the others because this signal is enriched from multiple cells during syncytium formation. Thus, this is a very sensitive assay to detect SARS2-S mediated syncytium formation from cell-cell fusion, which is blocked by chloroquine.

To avoid any further confusion, we have removed Fig.5I and its related text (after line 263), as suggested by this reviewer.

3. Based on the same concerns about the cell-cell fusion assay, this reader is also of the opinion that Fig. 4D should be deleted and that Lines 25-26 of the Abstract should be revised, for example to read: “NPC1 strongly promotes productive SARS-CoV-2 entry, which we propose is due to its influence on fusion in late endosomes.”

As suggested by this reviewer, we have deleted Fig.4D and its related text (after line 207) and revised the Abstract (line 26).

4. The authors should make a few more modifications to the Discussion: (a) Text between L323-329. Eliminate the terms “dominant” and “minor”; NPC1-independent entry through the plasma membrane in nasal cavity, where the extracellular pH was recorded to be ~6.7, is likely an/the most important route of infection clinically. (b) L347-348: specify that “in some cells” ACE2 is more likely an attachment factor.

“dominant” and “minor” have been removed, and these changes have been made in lines 316-324, and 342.

5. Line 154: Add to end of sentence: “, albeit higher than the IC50s recorded for inhibition of Ebola pseudovirion infections”.

This change has been made in line 152.

6. Line 217: Delete “strongly”; the effect of NPC1 KO on SARS2 infection of Caco2 cells is less than 1 log10.

It has been deleted in line 214.

7. Line 257: Replace “it” with “mildly low pH”.

It has been replaced in line 253.

8. Fig. 5I: Correct spelling of mCherry.

We apologize for this typo. Fig.5I has been deleted per this reviewer’s suggestion.

9. Line 292: The statement should be qualified as it looks like Omicron D18 S IP’d with the 1-377 fragment of NPC1 (in Supp. Fig. 12, IP lane 11).

The text has been revised accordingly in line 283.

10. L312: Specify the cell type for Ref 22 NPC1 KO, as cell type is critical to relative usage of plasma membrane vs. endosomal entry for SARS-CoVs. Ref. 22 used NPC1 KO 293T-

ACE2-TMPRSS2 cells.

It has been specified in line 304.

11. L351: Do you mean to say: “to initiate endocytic entry”. Alternatively, say “to promote Ebola particle internalization”.

It has been changed to “initiate” in line 346.

Reviewer #2 (Remarks to the Author):

The authors have addressed the concerns of the reviewers.

Thanks!

The small remaining comments are as follows:

1) lines 178, 219 and 317- because infection is not completely blocked NPC1 KO, it would be more appropriate to say NPC1 is involved (or a critical factor) rather than required which implies absolute.

As suggested by this reviewer, “required” has been changed to “a critical factor” in lines 178, 216, 264, 311, and several other places.

2) there are still English issues, but hopefully these will be addressed during editorial review.

We apologize that we also found a few other errors, which have been corrected.